# A fluorogenic probe for granzyme B enables in-biopsy evaluation and screening of response to anticancer immunotherapies

Jamie I. Scott[1], Lorena Mendive-Tapia[1], Doireann Gordon[1], Nicole D. Barth[1], Emily J. Thompson[1], Zhiming Cheng[1], David Taggart[1], Takanori Kitamura [2], Alberto Bravo-Blas[3], Edward W. Roberts [3], Jordi Juarez-Jimenez [4], Julien Michel[4], Berber Piet [5], I. Jolanda de Vries [6], Martijn Verdoes [6], John Dawson [7], Neil O. Carragher[7], Richard A. O' Connor[1], Ahsan R. Akram [1], Margaret Frame[7], Alan Serrels [1] & Marc Vendrell [1✉]

Immunotherapy promotes the attack of cancer cells by the immune system; however, it is difficult to detect early responses before changes in tumor size occur. Here, we report the rational design of a fluorogenic peptide able to detect picomolar concentrations of active granzyme B as a biomarker of immune-mediated anticancer action. Through a series of chemical iterations and molecular dynamics simulations, we synthesize a library of FRET peptides and identify probe **H5** with an optimal fit into granzyme B. We demonstrate that probe **H5** enables the real-time detection of T cell-mediated anticancer activity in mouse tumors and in tumors from lung cancer patients. Furthermore, we show image-based phenotypic screens, which reveal that the AKT kinase inhibitor AZD5363 shows immune-mediated anticancer activity. The reactivity of probe **H5** may enable the monitoring of early responses to anticancer treatments using tissue biopsies.

[1] Centre for Inflammation Research, Queen's Medical Research Institute, The University of Edinburgh, Edinburgh, UK. [2] MRC Centre for Reproductive Health, Queen's Medical Research Institute, The University of Edinburgh, Edinburgh, UK. [3] Cancer Research UK Beatson Institute, Glasgow, UK. [4] EaStChem School of Chemistry, Joseph Black Building, The University of Edinburgh, Edinburgh, UK. [5] Department of Pulmonary Diseases, Radboud University Medical Centre, Nijmegen, Netherlands. [6] Department of Tumor Immunology, Radboud Institute for Molecular Life Sciences, Radboud University Medical Centre, Nijmegen, Netherlands. [7] Cancer Research UK Edinburgh Centre, The University of Edinburgh, Edinburgh, UK. ✉email: marc.vendrell@ed.ac.uk

The response of CD8+ T cells is one of the main immune mechanisms to protect the human body against cancer. The presence of CD8+ T cells in tumors is indicative of a favorable prognosis in cancer patients[1–6]; however, there is high variability in how patients respond to immunotherapies[7–9]. Clinical imaging can measure the size of tumors and, to some extent, the infiltration of CD8+ T cells but they cannot provide readouts on how efficiently the immune system is responding against cancer cells. This limitation hinders the evaluation of drugs and the personalized optimization of immunotherapies to minimize off-target toxicity[10–13]. Chemical approaches that directly measure the activity of tumor-infiltrating T cells will accelerate the screening and optimization of anticancer drugs as well as improve the monitoring of early responses to treatment in a personalized manner.

Standard methods for in vitro monitoring of CD8+ T cell cytotoxicity employ lactate dehydrogenase (LDH), tetrazolium dye-based metabolic assays (MTS, MTT) or $^{51}$Cr release assays[14–16]. These assays report bulk cytotoxicity rather than T-cell-specific anticancer responses. The activity of CD8+ T cells can be indirectly monitored by measuring the concentration of extracellular cytokines[17] and membrane proteins (e.g., CD107a) using antibodies[18]. These methods assess T cell function but do not directly report on cancer cell death, thus are not useful biomarkers of the immune killing capacity in tumors. The chemical design of activity-sensing reporters of granzyme B (GzmB) represents a potentially effective strategy for monitoring the cytotoxic activity of CD8+ T cells in cancer. GzmB is a serine protease that is stored inactive in T cells until antigen-driven recognition prompts its release and activation inside cancer cells. Probes for detecting GzmB include antibodies and fusion proteins, but these do not distinguish between the active and inactive forms of the enzyme[19], and activatable constructs based on the Ile-Glu-Pro-Asp (IEPD) sequence first described by Thornberry et al.[20]. Some examples of the latter include poly-lysine graft copolymers for myocarditis[21], contrast agents for preclinical imaging[22–24], and nanoprobes for urinalysis[25,26]. These probes rely on a chemical scaffold with limited reactivity (e.g., $V_{max}$ in the range of pmol min$^{-1}$ and $k_{cat}/K_M$ ratios around mM$^{-1}$ s$^{-1}$, Table 1).

In this work, we rationally design GzmB substrates to identify peptide sequences with an optimized fit into the active site of the enzyme, suggesting that an alternative mode of binding can be exploited for the design of high-affinity probes for GzmB. Following development into FRET constructs we furnish probe H5, which exhibits $k_{cat}/K_M$ ratios several orders of magnitude greater than those observed for IEPD tetrapeptides. We then use probe H5 to enable real-time measurements of the anticancer activity of T cells in a mouse model of immune-mediated tumor regression. We also deploy H5 in image-based screens to identify drugs able to reinvigorate CD8+ T cell activity against cancer cells. Finally, we optimize the application of probe H5 in tissues from lung cancer patients to detect in situ T cell cytotoxic activity in human tumors.

## Results

**Design of a highly reactive fluorogenic probe for granzyme B.** The tetrapeptide IEPD has been the main scaffold reported for the preparation of covalent inhibitors as well as fluorogenic substrates targeting human GzmB (hGzmB)[22–29]. First, we assessed the reactivity of the commercial Ac-IEPD-AMC (i.e., a substrate that releases 7-amino-4-methylcoumarin upon reaction with hGzmB) against concentrations of an enzyme that would be applicable in clinically relevant assays. We observed slow enzymatic cleavage rates of Ac-IEPD-AMC (i.e., <1% cleavage at 20 nM enzyme for 2 h, 10% cleavage at 100 nM for 24 h) and a limit of detection (LoD) of 25 nM (Fig. S1), which greatly exceeds the pM concentrations of GzmB found in clinical samples[30,31]. In view of these results, we examined whether IEPD-based Förster Resonance Energy Transfer (FRET) probes[32–38] would show increased reactivity for hGzmB. We synthesized FRET substrates by flanking the IEPD sequence with fluorophores and quenchers as donor-acceptor pairs. Unlike in Ac-IEPD-AMC, we put the fluorophore at the N-terminal end to favor the trapping of the fluorescent peptide fragments resulting from the reaction with GzmB. Different combinations of fluorophores, spacers, and quenchers were synthesized, and the tetrapeptide T1 (with BODIPY-FL as the fluorophore and ethylenediamine-Dabcyl as the quencher) was identified as the most reactive substrate (Table S1). Still, the reactivity of T1 against hGzmB was poor (i.e., <5% cleavage at 20 nM enzyme for 2 h, 14% cleavage 100 nM for 24 h) with a LoD of 17 nM (Fig. S1, Table 1).

Therefore, we decided to optimize FRET substrates by identifying sequences that could react faster and more specifically with hGzmB. The IEPD core of the sequence was left untouched and instead we investigated the positions P1' and P2' of potential substrates because (1) C-terminal modifications of the IEPD sequence can be well tolerated by GzmB[39], and (2) proteomic studies revealed that hexapeptides can be good substrates for chymotrypsin-like and caspase-like proteases[40]. Therefore, we prepared FRET hexapeptides where the IEPD sequence was extended with small amino acids at the positions P1' and P2' (Fig. 1a), as suggested from substrate profiling in proteomic studies[41,42]. The hexapeptides H1 (IEPDAG) and H2 (IEPDSG) showed remarkably faster conversion than the tetrapeptide T1, although incomplete after 2 h with 20 nM hGzmB (e.g., 75 and 80% conversion, respectively, Table S2). Since the peptide H2 showed slightly higher reactivity, we retained serine in the P1' position and prepared the peptide H3 (IEPDSL) with a more hydrophobic leucine in the P2' position. This change did not improve the reactivity (e.g., 80% conversion, Table S2) so we explored alternatives in the P1' position, including large (i.e., tryptophan, H4: IEPDWL) and small hydrophobic amino acids (i.e., alanine, H5: IEPDAL)[43]. The peptide H4 achieved 82%

**Table 1 Summary of fluorescent peptide-based probes for GzmB.**

| Compound | Amplification mechanism | $K_{cat}$ (s$^{-1}$) | $K_M$ ($\mu$M) | $K_{cat}/K_M$ (M$^{-1}$ s$^{-1}$) | LoD | Ref. |
|---|---|---|---|---|---|---|
| **4-mer** | | | | | | |
| Ac-IEPD-AMC | Fluorogenic dye | 0.5 | 160 | $3.3 \times 10^3$ | 25 nM | Ref. [39] |
| PEGylated IONPs | Nanoconstruct | ~0.002 | ~0.2 | $1.1 \times 10^4$ | n.d. | Ref. [25] |
| CyGbP$_F$ | Fluorogenic dye | 0.07 | 22.7 | $3.1 \times 10^3$ | n.d. | Ref. [26] |
| T1 | FRET | 2.3 | 30.8 | $5.9 \times 10^4$ | 17 nM | This work |
| **6-mer** | | | | | | |
| H5 | FRET | 117 | 9.6 | $1.2 \times 10^7$ | 6 pM | This work |
| **9-mer** | | | | | | |
| qTJ71 | FRET | 0.65 | 9.9 | $6.6 \times 10^5$ | n.d. | Ref. [27] |

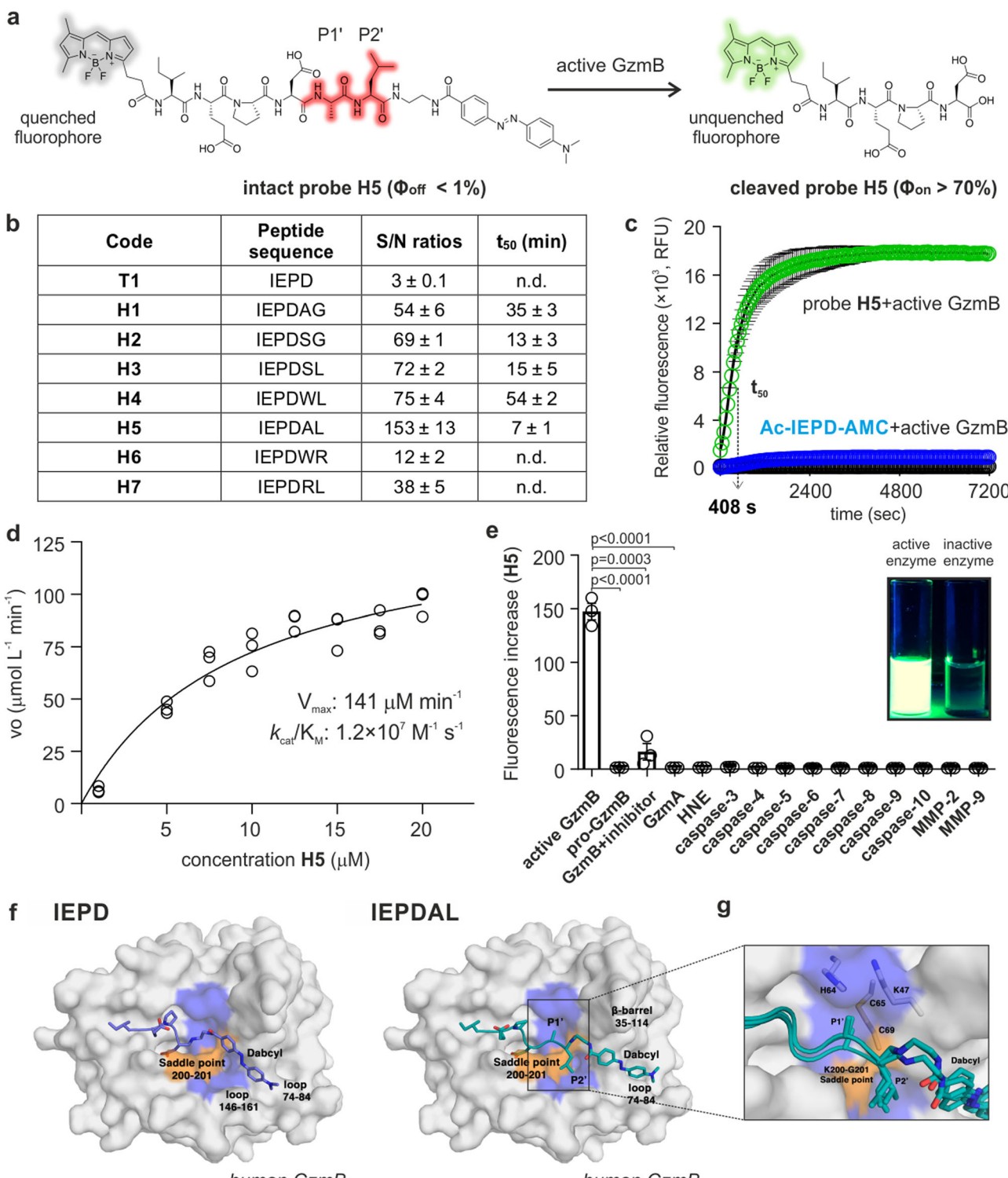

**Fig. 1 The hexapeptide H5 achieves high reactivity and selectivity for GzmB by accessing a unique binding pocket. a** Fluorogenic hexapeptide **H5** and fluorescence quantum yields of intact and cleaved probe. **b** Tetra- and hexapeptide sequences, fluorescence increases and $t_{50}$ values upon incubation (25 μM) with hGzmB (20 nM). Data as means ± SEM ($n = 3$). **c** Time-course fluorescence of **H5** (25 μM, green, 510 nm) and Ac-IEPD-AMC (25 μM, blue, 450 nm) after incubation with hGzmB (20 nM) at 37 °C. Probe **H5** alone (25 μM, black, 510 nm). Data as means ± SEM ($n = 5$). **d** Cleavage rate of probe **H5** by hGzmB (20 nM) as a function of substrate concentration. Data as individual replicates and kinetic values determined using the Michaelis-Menten equation ($n = 3$). **e** Fluorescence changes of probe **H5** (25 μM) after incubation with proteases (20 nM) at 37 °C for 60 min. Data as means ± SEM ($n = 3$). For active GzmB vs pro-GzmB, $p < 0.0001$. For active GzmB vs GzmB+ inhibitor, $p = 0.0003$. For active GzmB vs GzmA, $p < 0.0001$. **f** Representative binding mode of IEPD-Dabcyl in **T1** (left) and IEPDAL-Dabcyl in **H5** (right) from the MD simulations. **g** Detailed interactions at P1′ and P2′ sites for the probe **H5** with overlaid structures from 3 independent runs. *P*-values from two-tailed *t*-tests. Source data (**b**–**e**) provided as a Source Data file.

conversion whereas the peptide **H5** showed the highest reactivity, with fast and complete cleavage in less than 30 min (Table S2 and Fig. S2). Interestingly, this result suggested that a smaller amino acid (i.e., alanine instead of tryptophan) was better tolerated by the enzyme. To confirm this hypothesis, we synthesized negative control peptides of the **H4** and **H5** sequences where we included an arginine either in P1' or P2', given that the hGzmB crystal structure features one arginine (Arg226) in the S1 subsite and electrostatic repulsion between arginine residues should disfavor binding[43]. The resulting hexapeptides **H6** (IEPDWR) and **H7** (IEPDRL) showed markedly reduced reactivity (i.e., 40 and 60% conversion, respectively, Table S2), yet still much higher reactivity than the tetrapeptide **T1**. These results suggest that FRET hexapeptide constructs react with hGzmB much faster than shorter tetrapeptides. We prepared all peptides using solution and solid-phase synthesis (Supplementary Note 1), isolated them by preparative HPLC in purities over 95%, and confirmed their identity by high-resolution mass spectrometry (Table S3).

**Probe H5 is a highly specific substrate of human GzmB by accessing a unique binding pocket.** We analyzed the fluorogenic response of all hexapeptides **H1-H7** by measuring their fluorescence emission in the presence of recombinant hGzmB (Fig. 1a). As expected, all hexapeptides outperformed the tetrapeptide **T1**, with signal-to-background ratios ranging from 12-fold (for peptide **H6**) to 153-fold (for peptide **H5**) (Fig. 1b and Fig. S3). We also measured the time taken for the most reactive compounds to reach 50-fold fluorescence increase (termed $t_{50}$). Probe **H5** showed the fastest response among all peptides with a $t_{50}$ of 7 min (Fig. 1b, c). We also compared the kinetic properties of probe **H5** to previously reported tetrapeptide constructs and the best three hexapeptides (**H2**, **H3**, and **H4**, Table S4). The probe **H5** showed remarkably high catalytic efficiency with $V_{max}$ values in the high μM min$^{-1}$ range and a $k_{cat}/K_M$ ratio of $1.2 \times 10^7$ M$^{-1}$s$^{-1}$ (Fig. 1d), which represents more than 1000-fold improvement over fluorescent tetrapeptides (Table 1). These results confirm the exceptional reactivity of probe **H5**, which achieves an unprecedented LoD for hGzmB of 6 pM without any washing steps (Fig. S4).

We also assessed the selectivity of the probe **H5** for hGzmB over other enzymes, including the inactive pro-hGzmB and other proteases that are active in tumors or during apoptosis (e.g., caspases, granzymes, matrix metalloproteinases (MMPs), and neutrophil elastase). Probe **H5** showed excellent specificity for hGzmB with minimal response to other enzymes (Fig. 1e), including MMPs and caspases, which can cleave some IEPD-based probes (Fig. S4). This represents one important advantage when assessing response to immunotherapy, because the cross-reactivity with caspase-3 would impede distinguishing generic cell death from T-cell-mediated cancer cell death. Finally, to further confirm that the fluorescence generated from probe **H5** was due to its specific reaction with hGzmB, we performed experiments in the presence of the aldehyde-tagged reversible GzmB inhibitor Ac-IEPD-CHO, which markedly reduced the fluorescence response of **H5** (Fig. 1e).

In order to understand the differences in reactivity between the model tetrapeptide **T1** and our hexapeptides, we compared the preferred mode of binding of **T1** and **H5** in complex with a model of hGzmB using molecular dynamics (MD) simulations. Because all hexapeptides shared the BODIPY-FL and P1-P4 amino acids, we focused our analysis on the P1'-P2' residues and the Dabcyl quencher. The MD simulations revealed that **T1** and **H5** accommodated the quencher in two distinct pockets. In the hGmzB-**T1** simulations, the Dabcyl moiety bound preferentially to the cleft between the loop 74–84 and the loop 146–161

(Fig. 1f), whereas in the hGmzB-**H5** simulations, the quencher bound preferentially to the extension of the catalytic cleft delineated by the loop 74–84 and the β-barrel 35–114 (Fig. 1f). Visual inspection of the trajectories revealed that the backbone of residues 200–201 creates a saddle point—highlighted in orange in Fig. 1f—that precludes the shorter tetrapeptide **T1** from accommodating the quencher along the catalytic cleft. In both cases, the Dabcyl moiety remains flexible and does not form long-lived interactions with specific protein residues. The MD simulations suggest the enhanced reactivity of **H5** over the other hexapeptides is due to the combined effects of positioning an alanine residue in P1', which fills a small hydrophobic pocket in the catalytic cleft, and a leucine residue in P2', which straps the 200–201 saddle point (Fig. 1g). This binding mode fulfills the complementarity criteria that drive the formation of densely packed hydrophobic structures, so-called steric zippers, found in prion proteins[44]. Moreover, it has been recently shown that mutations of residues that disrupt knobs-into-holes packing dramatically reduce the binding affinity of short peptides[45]. Therefore, the proposed binding mode explains the enhanced reactivity of probe **H5** over: (1) peptides **H4** and **H6**, because tryptophan in P1' is too large to fit in the pocket, (2) peptide **H7**, whose arginine in P1' is disfavored due to electrostatic interactions with the basic residues that delineate this sub-pocket, (3) peptide **H3**, whose serine in P1' lacks neighboring effective hydrogen-bond donors, and (4) peptides **H1** and **H2** because the glycine in P2' is too small to pack effectively against the 200–201 saddle point.

**Probe H5 detects real-time reinvigoration of T cells in co-cultures with cancer cells.** Given the reactivity and selectivity of probe **H5** for GzmB, we studied its utility to measure the cytotoxic activity of T cells during the attack on cancer cells. To investigate this, we co-cultured mouse CD8+ T cells and E0771 mammary tumor cells (Fig. 2a). Substrates for mouse GzmB feature phenylalanine in P2—instead of proline in human substrates[46]—, yet we observed a good response of the probe **H5** to mouse GzmB, with slightly better reactivity than the hexapeptide analogue containing phenylalanine (probe **H5$_m$**: IEF-DAL, Fig. S5). Therefore, we decided to use probe **H5** for both mouse and human assays. First, we optimized the co-cultures to effectively increase the levels of GzmB in CD8+ T cells. Incubation with interleukin IL-2, an essential cytokine for the survival, proliferation, and activation of CD8$^+$ T cells, produced the largest reinvigoration, with over 80% CD8+ T cells expressing GzmB after IL-2 treatment (Fig. S6) and confirmation by fluorescence microscopy using anti-GzmB (Fig. 2b). Next, we applied these conditions to co-cultures of CD8+ T cells and genetically modified E0771-La2-NLR cells, which express the red fluorescent protein mKate[47] to facilitate their detection and incubated them with probe **H5** and Annexin V-AF647, a marker of apoptosis. Flow cytometric analysis revealed that >80% of cancer cells that had been cultured with activated T cells were double-stained with **H5** and Annexin V-AF647, confirming that the emission of probe **H5** reports immune-mediated cancer cell death (Fig. 2c and Fig. S7). Co-cultures of cancer cells with non-reinvigorated T cells or cancer cells treated with staurosporine—a relatively non-specific kinase inhibitor that induces apoptosis—showed weak fluorescence, confirming that probe **H5** only detects dead cancer cells that have been killed by CD8+ T cells and not all dead cancer cells (Fig. 2c and Fig. S7). Furthermore, we observed that the fluorescence signals of probe **H5** were significantly reduced upon blockage with the GzmB inhibitor Ac-IEPD-CHO (Fig. S7). Microscopy experiments also corroborated that probe **H5** only stained the cytoplasm of mKate+ cancer cells, but not T cells

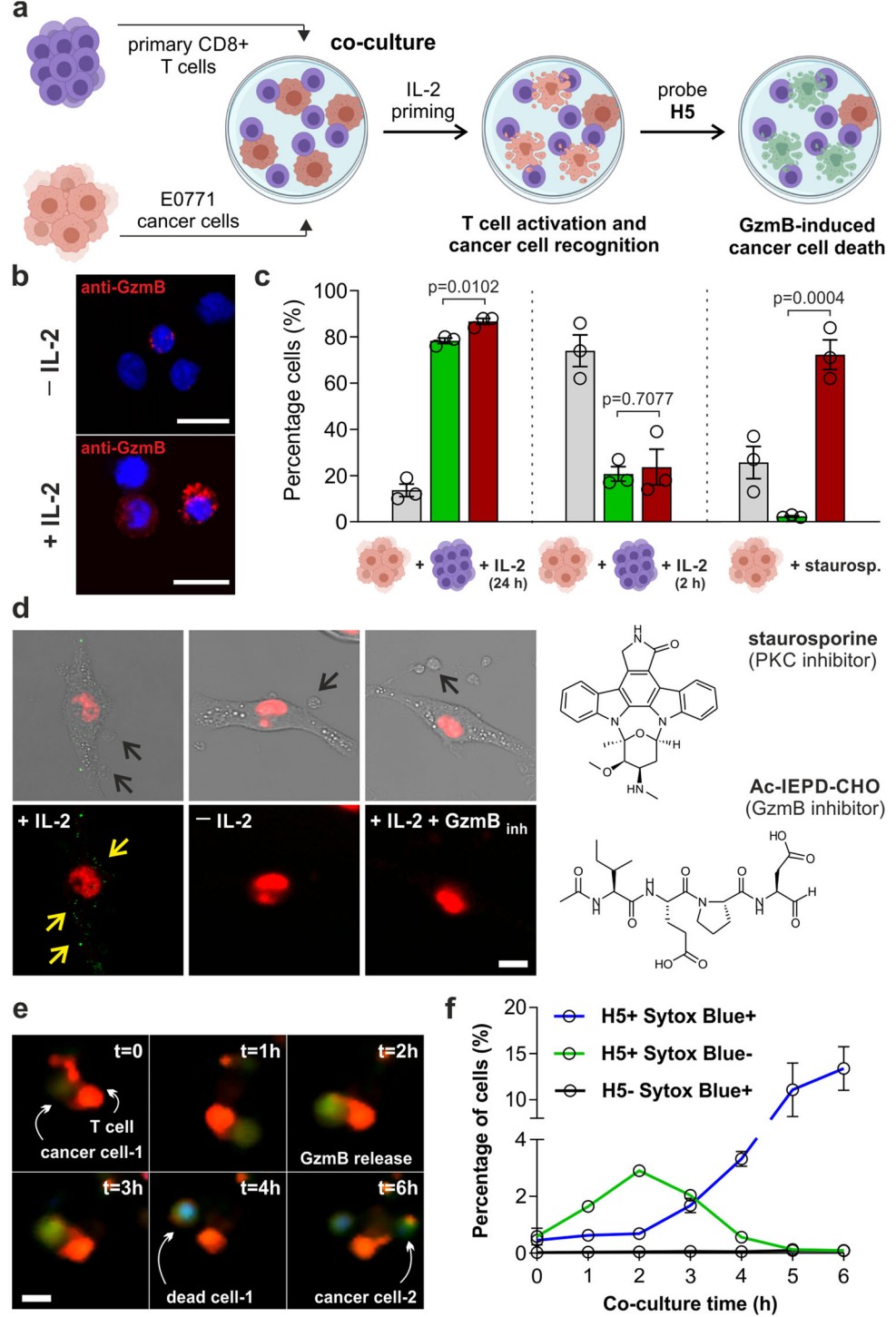

**Fig. 2 Probe H5 detects GzmB-mediated anticancer activity of CD8+ T cells. a** Schematic procedure for co-culture assays. **b** Representative microscopy images of CD8+ T cells before/after reinvigoration and staining by anti-GzmB (10 nM, red) and Hoechst 33342 (1 µM, blue) ($n = 3$). Scale bar: 10 µm. **c** Flow cytometry (gating: Fig. S7) of E0771 cells in co-culture with active (IL-2, 24 h) or inactive (IL-2, 2 h) CD8+ T cells, or alone with staurosporine (1 µM, 1 h). Legends: viable (gray), **H5**-stained (green), apoptotic (red). **H5** (5 µM), Annexin V-AF647 (10 nM). Data as means ± SEM ($n = 3$). **d** Confocal microscopy of mKate-expressing E0771 cancer cells (red) stained with **H5** (25 µM, green) in co-culture with active T cells (left), non-active T cells (centre), or active T cells plus Ac-IEPD-CHO (right). Black arrows highlight T cells, yellow arrows highlight **H5**-stained intracellular GzmB puncta. Quantification of fluorescence intensity by image analysis shown in Fig. S8. Scale bar: 10 µm. **e** Time-course fluorescence microscopy of OT-I CD8+ T cells stained with Cell Tracker Orange (red) killing OVA-EL4 cancer cells in the presence of **H5** (10 µM, green) and Sytox Blue (1 µM, blue) ($n = 3$). Scale bar: 10 µm. **f** Flow cytometric analysis of OVA-EL4 cancer cells from experiments in **e**. Data as means ± SEM ($n = 3$). *P*-values from two-tailed *t*-tests. Source data (**c**, **f**) provided as a Source Data file.

(Fig. 2d and Fig. S8). As additional controls, we confirmed that probe **H5** was not cytotoxic (Fig. S9) and did not stain cancer cells in co-cultures where T cells were inactive or in co-cultures that had been pretreated with the reversible GzmB inhibitor Ac-IEPD-CHO (Fig. 2d). Furthermore, we synthesized an 'always-on' derivative of probe **H5** (**H5**-unquenched) to confirm its permeability in different cancer cell lines (Fig. S10). Altogether, these results confirm that probe **H5** can detect cancer cell death resulting from the attack of invigorated CD8+ T cells and suggests that the fluorescence emission of probe **H5** can be used as a biomarker of immunomodulatory efficacy in live cultures.

Next, we examined whether **H5** could be used to image how CD8+ T cells attack cancer cells in real-time with an antigen-specific method of T cell activation. We utilized CD8+ T cells expressing the OT-I transgenic receptor, which specifically targets cells presenting the OVA-derived SIINFEKL antigen in the context of H-2kb and co-cultured them with SIINFEKL-pulsed EL4 cancer cells (Fig. S11). T cells were counterstained with Cell Tracker Orange for ease of identification and incubated with probe **H5** and the cell death marker Sytox Blue before time-lapse microscopy. Initially, the probe **H5** was silent but its emission inside target cells gradually increased as CD8+ T cells started to form immune synapses with cancer cells (Fig. 2e and Movie S1 and S2). After several contacts between CD8+ T cells and cancer cells, the green fluorescence signal of **H5** colocalized with the blue signal of Sytox Blue, indicating that targeted cancer cells had undergone GzmB-mediated apoptosis. Quantitative flow cytometric analysis also confirmed that the fluorescence signal of the probe **H5** in cancer cells preceded that of Sytox Blue, which corroborates active GzmB as an early biomarker of immune-mediated cancer cell death (Fig. 2f). In parallel, we performed in vitro experiments mimicking aspects of the intracellular environments found in cancer cells that receive multiple contacts —and thus multiple deliveries of GzmB—from CD8+ T cells[48] and observed an accumulative fluorogenic signal for probe **H5** to spiked 1 nM increments of GzmB (Fig. S12).

**Probe H5 detects T-cell-mediated tumor regression in a mouse model of cancer.** We further investigated the capability of the probe **H5** to detect GzmB activity in a mouse model of tumor regression. Serrels et al reported that inhibition of Focal Adhesion Kinase (FAK) can drive T-cell-mediated regression of squamous cell carcinoma (SCC) tumors via modulation of the immuno-suppressive microenvironment (Fig. 3a, b), with FAK inhibitors currently tested in clinical trials in combination with immune checkpoint inhibitors[49]. Because the tumor regression in SCC FAK (−/−) mice is dependent on CD8 T+ cells, this preclinical model represented an excellent platform to examine whether the probe **H5** could detect T-cell-mediated cancer cell death in tumors.

First, FVB immune-competent mice were challenged with SCC FAK (−/−) cancer cells or wild-type SCC cancer cells (as a negative control), and tumors were grown for 2 weeks. As expected, SCC FAK (−/−) tumors were significantly smaller and contained fewer live cancer cells (Fig. 3c) than wild-type SCC tumors. The number of CD8+ T cells was around 10-fold higher in SCC FAK (−/−) tumors, implying an invigorated T cell response (Fig. 3d), which also showed higher expression levels of GzmB as determined by ELISA (Fig. S13). In order to evaluate whether the cancer cells in SCC FAK (−/−) tumors contained intracellular active GzmB, tumors were harvested and treated with probe **H5** for 30 min before being analyzed by flow cytometry. Notably, over 40% of cancer cells in the SCC FAK (−/−) tumors were labeled with probe **H5**, whereas wild-type SCC tumors were almost not stained (Fig. 3e, f). Furthermore, we

observed that the fluorescence signal of the probe **H5** was exclusively found in a subset of cancer cells but not in other cells found in tumors (e.g., monocytes, fibroblasts, CD4 + T cells) (Fig. 3g, h and Fig. S14). Altogether, these results corroborate the utility of probe **H5** to rapidly detect T-cell-mediated cell death in mouse tumors.

**Probe H5 identifies immunomodulatory activity in drug screens and human tumors from lung cancer patients.** We next assessed the utility of the probe **H5** in screens of immunomodulatory drugs and clinical assays in human tumors. First, we adapted our co-cultures of mouse CD8+ T cells and E0771 cancer cells to a 384-well plate format for image-based phenotypic assays that could screen small molecules for invigorating the killing capacity of CD8+ T cells (Fig. 4a). We used the ImageXpress high-content analysis system to test a collection of anticancer drugs with varied pharmacological targets (Table S5). E0771 and CD8+ T cells were co-cultured for 2 days and then incubated with IL-2 (100 U mL$^{-1}$) and each individual drug at their respective working concentrations for 3 h (Table S5). One hour before imaging, the probe **H5** was added to the wells and we acquired fluorescence microscopy images, including Hoechst 33342 as a nuclear counterstain. The fluorescence intensity of probe **H5** inside cancer cells was used to compare the immunomodulatory capacity of all 44 drugs. We included wells with a high concentration of IL-2 (250 U mL$^{-1}$) as a positive control for high GzmB and wells with 100 U mL$^{-1}$ IL-2 plus rapamycin, a known mTOR inhibitor that blocks IL-2-induced activation of T cells, as a negative control for low GzmB activity. Of note, a small set of compounds with different pharmacological functions (e.g., protein kinase inhibitors, inhibitors of microtubule function, DNA alkylating agents, proteasome inhibitors) exhibited superior staining than the single treatment with IL-2, indicating the ability to invigorate anticancer T cell activity (Fig. 4b; for an extended description of the screening results, see Supplementary Note 2). Among these, the AKT kinase inhibitor AZD5363 (1 μM) exhibited the brightest **H5** fluorescence staining and was selected for further studies. We analyzed the cell viability and extent of **H5** fluorescence staining in E0771 cancer cells that had been incubated with AZD5363 as well as in CD8+ T cell plus E0771 cell co-cultures that were incubated with IL-2 only or IL-2 plus AZD5363. The treatment of AZD5363 on its own did not induce significant cancer cell death or led to brighter fluorescence labeling by the compound **H5**, whereas the same concentration of AZD5363 in combination with 100 U mL$^{-1}$ IL-2 caused significant cancer cell death and **H5** fluorescence emission in CD8+ T cell-E0771 cell co-cultures (Fig. S16). These observations are in agreement with recent reports that suggest AKT inhibition may prevent CD8+ T cell exhaustion, resulting in enhanced cytolytic activity against target cells[50]. These results highlighted the potential of AZD5363 to invigorate the anticancer activity of CD8+ T cells when used in combination with IL-2 and demonstrate the utility of our **H5**-based imaging screen platform for the identification of immunomodulatory drugs.

Finally, we examined whether probe **H5** could also monitor cytotoxic T cell function in biopsies from lung cancer patients as a potential method to screen their predisposition to anticancer treatments. For these experiments, we obtained paired (i.e., tumor and non-tumor, representative images displayed in Fig. 4c) tissue resections from five treatment-naïve lung cancer patients undergoing surgical resection. To analyze whether probe **H5** could detect ongoing GzmB-mediated killing of tumor cells, we treated all samples with probe **H5** (30 min, r.t.) and surface markers for epithelial cells (EpCAM), leukocytes (CD45), and cytotoxic T lymphocytes (CD8) before flow cytometry analysis. As shown in

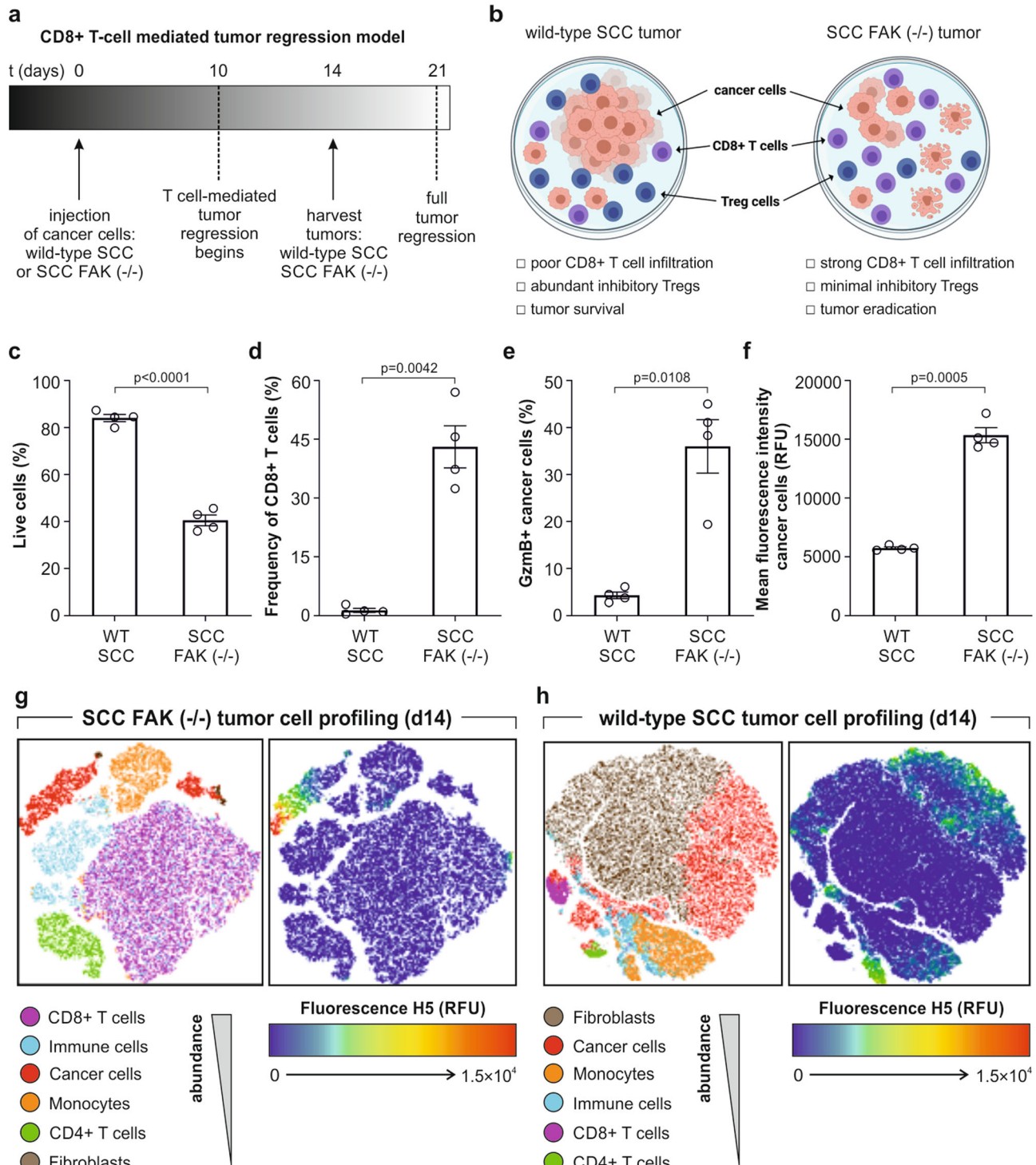

**Fig. 3 Probe H5 detects T-cell-mediated tumor regression in a mouse model of squamous cell carcinoma. a** Experimental timeline of the CD8+ T-cell-mediated tumor regression model. **b** Cell populations found in wild-type SCC and SCC FAK (−/−) tumors. **c–f** SCC and SCC FAK (−/−) cells were injected into FVB immunocompetent mice (1 × 10⁶ cells/mouse) and tumors were harvested on day 14. Flow cytometry (gating: Fig. S15) of wild-type SCC and SCC FAK (−/−) tumors for cell viability with live/dead stain (**c**); CD8+ T cell infiltrates with anti-CD8-PE (p < 0.0001) (**d**); percentage of GzmB-positive SCC cancer cells by staining with **H5** (5 µM, 30 min) (p = 0.0042) (**e**); fluorescence intensity of **H5** inside SCC cancer cells (525 nm) (p = 0.0108) (**f**). Data in **c–f** as means ± SEM (n = 4) (p = 0.0005). **g, h** SCC FAK (−/−) tumors (**g**) and wild-type SCC tumors (**h**) (ex vivo stained with 5 µM compound **H5**) were analyzed by flow cytometry and presented as pseudo-colored two-dimensional tSNE (t-distributed stochastic neighbor embedding) plots to determine the distribution of the probe in different cell populations (left) and the fluorescence intensity of **H5** staining (right) (n = 4). P-values from two-tailed t-tests using Welch correction. Source data (**c–f**) provided as a Source Data file.

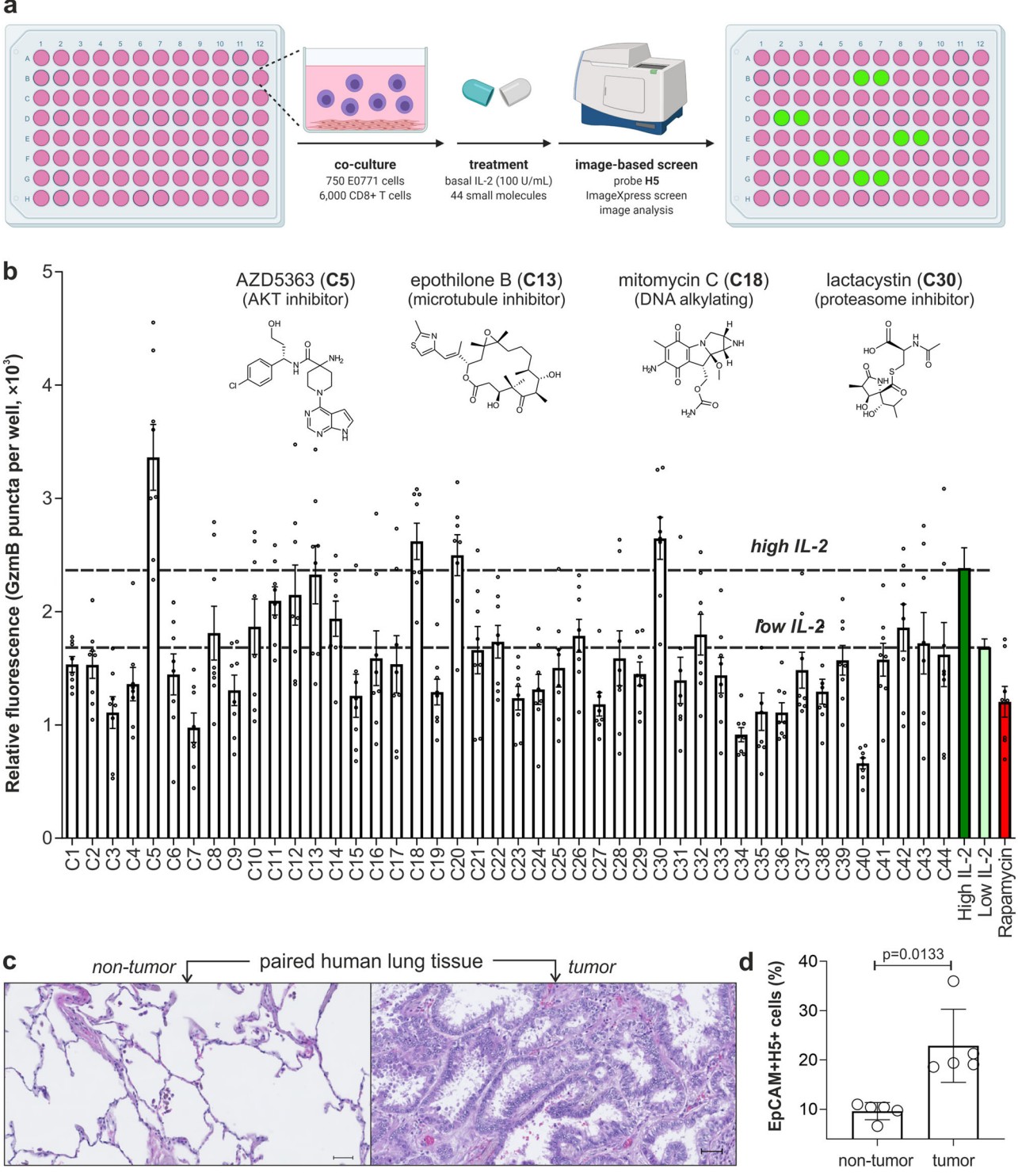

**Fig. 4 Probe H5 detects immunomodulatory action in phenotypic screens and T cell cytotoxic activity in human tissue from lung cancer patients.**
**a** Experimental protocol of the phenotypic screen. **b** Fluorescence intensity of probe **H5** in co-cultures of E0771 cells and IL-2-activated CD8+ T cells after incubation with small molecules (C1-C44). Wells received 100 U mL$^{-1}$ IL-2 (low IL-2) for T cell viability. High IL-2 (250 U mL$^{-1}$) used as a positive control for invigorated CD8+ T cells and rapamycin (0.3 μM) used as a negative control. Probe **H5** (20 μM) and Hoechst 33342 (1 μM) were incubated for 1 h and fluorescence images were acquired with an ImageXpress$^{TM}$ XLS. Data as means ± SEM ($n = 8$). Chemical structures of drugs showing **H5** fluorescence signals above those with 250 U mL$^{-1}$ IL-2. **c** Representative H&E microscope images of non-tumor (left) and lung adenocarcinoma (right) paired samples. Scale bar: 50 μm. **d** Cytometry analysis (gating: Fig. S17) showing the percentage of EpCAM+ **H5**+ epithelial cells found in paired tissues (non-tumor vs tumor) from lung cancer patients after incubation with probe **H5** (5 μM). Data as means ± SD ($n = 5$). *P*-value from two-tailed *t*-tests. Source data (**b**, **d**) provided as a Source Data file.

Fig. 4d, we observed a significant increase in the percentages of probe **H5**+ cells within the EpCAM + population in cancer tissue across all patients, indicative of the cytotoxic activity of CD8+ T cells against lung epithelial cells in tumors. The short excitation and emission wavelengths of BODIPY-FL might be limiting for some in vivo imaging studies, therefore further optimization studies will be needed to adapt probe **H5** for the evaluation of immunotherapy regimes in cancer patients. Altogether, our results indicate that probe **H5** can be used in vitro in high-throughput screening assays to facilitate the discovery of immunotherapy combinations as well as in the clinical characterization of human tumor biopsies, opening new avenues to accelerate the development of personalized anticancer immunotherapies.

## Discussion

Fluorogenic peptides are excellent scaffolds to develop agents monitoring protease activity. We here describe the rational design of a fluorogenic probe for rapid detection and imaging of active mouse and human GzmB, a key protease involved in initiating apoptosis of infected or malignant cells. Starting from the generic IEPD sequence, we have built a collection of FRET probes to optimize the hexapeptide **H5** with an unprecedented $k_{cat}/K_M$ ratio of $1.2 \times 10^7$ $M^{-1}$ $s^{-1}$ and a limit of detection of 6 pM. Probe **H5** displays a limit of detection within the physiological levels of GzmB found in human tissues and fluids (e.g., serum, synovial fluid), therefore it holds potential for the diagnosis and monitoring of inflammatory diseases (e.g., arthritis or inflammatory bowel diseases, among others)[51–53]. Furthermore, we have used molecular dynamic simulations to investigate the binding modes of the peptides and observed that the probe **H5** binds to an extension of the catalytic cleft delineated by the loop 74–84 and the β-barrel 35–114 of hGzmB, which is not accessible by tetrapeptide sequences. The discovery of this alternative binding mode to active hGzmB may facilitate the design and optimization of future imaging probes and enzyme inhibitors, which could be key in combatting some of the damaging autoimmune functions of GzmB[51]. We have demonstrated that the fluorescence emission of **H5** can be used as a direct reporter of immune-mediated tumor-killing ability in live cultures of CD8+ T cells and cancer cells, both qualitatively by fluorescence microscopy and quantitatively by flow cytometry. Importantly, the probe **H5** does not fluoresce inside T cells, where GzmB is inactive, or in cancer cells when killed by other agents (i.e., staurosporine) that are not related to immune-mediated cancer cell death. This selectivity profile of probe **H5** may find application for monitoring GzmB activity in other diseases where the activation state of immune cells is currently under investigation, such as Epstein-Barr virus-mediated multiple sclerosis and chronic obstructive pulmonary disease[54,55]. We have also shown that the probe **H5** can identify mouse tumors undergoing immune-mediated regression in a model of squamous cell carcinoma. This feature may enable future rapid analyses of immunity in solid tumors, as tumor growth and immune infiltration might be difficult to distinguish by some clinical imaging modalities. Probe **H5** will also create avenues to design discovery platforms for identifying small molecule drugs invigorating immunomodulatory responses in image-based phenotypic screens. Finally, we have demonstrated the use of probe **H5** for the clinical analysis of human tumor biopsies, highlighting a potential application for the personalized detection of early responses to anticancer immunotherapies.

## Methods

**Chemical synthesis**. Full details of synthetic procedures and chemical characterization are included in the Supplementary Information.

**Fluorescence assays with recombinant enzymes**. Fluorescence-based assays with different enzymes were performed in the appropriate buffers. Probes (25 μM) were added to enzymes (20 nM or at the indicated concentrations) in 384-well plates and their fluorescence emission was recorded at 450 nm (for AMC) or 510 nm (for BODIPY) at 37 °C using a Synergy H1 Hybrid spectrophotometer. In experiments with Ac-IEPD-CHO (50 μM), the inhibitor was preincubated for 1 h with GzmB before the addition of any probe.

**Primary cell isolation and cell culture**. Highly metastatic derivative of E0771 mouse mammary tumor cells expressing the nuclear-localized red fluorescent protein mKate (obtained from Dr Takanori Kitamura and Professor Jeffrey Pollard at The University of Edinburgh) were grown using Dulbecco's Modified Eagle Medium (DMEM) supplemented with 10% fetal bovine serum (FBS), antibiotics (100 U mL$^{-1}$ penicillin and 100 mg mL$^{-1}$ streptomycin), and 2 mM L-glutamine in a humidified atmosphere at 37 °C with 5% $CO_2$[45]. Cells were regularly passaged in T-25 cell culture flasks upon reaching 90% confluency. Primary CD8+ T cells were isolated from mouse spleens (Strain: C57BL/6, Sex: female, Age: 10–14 weeks, obtained from Charles Rivers) by tissue homogenization, erythrocyte lysis, and purification with magnetic beads (CD8 Microbeads Kit).

**Computational studies and molecular dynamics simulations**. Models were built with Maestro and AMBER16 using PDB id. 1IAU[43]. The forcefields ff14SB, GAFF2, and TIP3P were used to parameterize the solvated systems. Torsional parameters for the azobenzene moiety in Dabcyl were provided by the Luque research group[56]. For each system, three replicates were run at 298 K and 1 atm for 200 ns using a Langevin thermostat (collision frequency of 3 ps$^{-1}$), and a Monte Carlo barostat was implemented in the CUDA accelerated version PMEMD[57]. Additional details are provided in the Supplementary Information.

**Flow cytometry**. E0771 cells ($5 \times 10^4$ cells/well) were plated on Geltrex-coated six-well plates with IL-2 (1000 U mL$^{-1}$) and co-cultured with either murine CD8+ T cells ($2.5 \times 10^5$ cells/well) which had been previously cultured for 2 days in E-DMEM with anti-mouse CD3e (2 μg mL$^{-1}$, ThermoFisher, Catalog: 16-0031-86), anti-mouse CD28 (5 μL mL$^{-1}$, ThermoFisher, Catalog: 16-0281-86), and IL-2 (80 U mL$^{-1}$). Treatments with staurosporine (1 μM) were for 1 h. Flow cytometry preparation included treatment with trypsin-EDTA (0.05%), wash, resuspension in PBS and incubation with probe **H5** (5 μM) for 1 h at 37 °C. Cells were then washed twice and incubated with Annexin V-AF647 (10 nM, 1:100 dilution) prior to flow cytometry analysis in a 5 L LSR (software: BD FACSDIVA V8.0) with data analyzed with FlowJo V10. Excitation/emission wavelengths: **H5** (488 nm, 525 ± 50 nm), mKate (561 nm, 635 ± 15 nm), Annexin V-AF647 (640 nm, 670 ± 14 nm).

**Live-cell fluorescence confocal microscopy**. E0771 ($1.5 \times 10^3$ cells/well) were plated on Geltrex-coated 8-chamber glass slides with freshly isolated murine CD8+ T cells ($1.2 \times 10^4$ cells/well) in enriched E-DMEM phenol red-free media with IL-2 (80 U mL$^{-1}$), anti-mouse CD3e (2 μg mL$^{-1}$, ThermoFisher, Catalog: 16-0031-86) and anti-mouse CD28 (5 μL mL$^{-1}$, ThermoFisher, Catalog: 16-0281-86). After 48 h, T cell reinvigoration was performed by the addition of IL-2 (1000 U mL$^{-1}$) and incubation for 3 h. Probe **H5** was used at 25 μM and Ac-IEPD-CHO at 50 μM with 1 h preincubation. After 1 h treatment at 37 °C with **H5**, cells were washed with media and imaged under a Leica TCS SP8 fluorescent confocal microscope (software: Leica application Suite X) under a ×40 oil objective. Excitation wavelengths: 488 nm (**H5**), 561 nm (mKate). Images were analyzed with ImageJ.

**Preclinical cancer model**. All animal work was carried out in compliance with UK Home Office guidelines. SCC tumors were established and labeled for flow cytometry as previously described[58]. Briefly, $1 \times 10^6$ SCC cancer cells (obtained from Dr Alan Serrels at The University of Edinburgh) were injected subcutaneously into FVB/N mice ($n = 4$ for both WT and FAK (−/−) groups, Sex: female, Age: 10–14 weeks, obtained from Harlan UK), which were sacrificed 14 days post-implantation and tumors disaggregated to generate a single-cell suspension. Cells were labeled with antibodies (1:100 dilution) and the probe **H5** for 30 min at r.t. and analyzed by flow cytometry. Data analysis was performed using FlowJo V10 software.

**Drug screening**. E0771 (750 cells/well) and freshly isolated murine CD8+ T cells (6000 cells/well) were plated in 384-well plates as described above. After 2 days of co-culture at 37 °C, cells were further incubated with IL-2 alone (100 or 250 U mL$^{-1}$) or IL-2 (100 U mL$^{-1}$) in combination with drugs (Table S5) for 3 h. Compound **H5** (20 μM) and Hoechst 33342 (1 μM) were added for 1 h at 37 °C, cells were washed with PBS and imaged under ImageXpress$^{TM}$ XLS using KRB (10% FBS) buffer as media. Fluorescence images (4 sites/well) were acquired under a ×40 objective using 405 nm (Hoechst) and 488 nm (**H5**) excitation wavelengths and data were analyzed with MetaXpress 3 software using the Custom Module Editor. Briefly, nuclei were used as seeds to create a pseudo-cell area and GzmB-positive puncta were detected using a top-hat filter (pixel size: 10, circle shape) followed by an average filter ($3 \times 3$ pixels).

**Human tumors**. Tissue samples of tumor and paired non-tumor ($n = 5$ for both conditions) lungs were taken from treatment-naïve patients undergoing surgical resection. Participants provided written informed consent and the study was approved by NHS Lothian REC and facilitated by NHS Lothian SAHSC Bioresource (REC No: 15/ES/0094). Fresh samples were minced, incubated with $1\ mg\ mL^{-1}$ collagenase IV, $1\ mg\ mL^{-1}$ DNase 1, $50\ U\ mL^{-1}$ hyaluronidase in DMEM for 1 h at 37 °C with agitation, and passed through 100 and 70 μm filters followed by red cell lysis. Single-cell suspensions were washed in PBS and stained with Zombie UV prior to surface staining for EpCAM-BV650 (1:100, BioLegend, Catalog: 324226), CD45-APCcy7 (1:100, BD, Catalog: 557833), and CD8-BV421 (1:100, BioLegend, Catalog: 344748) for 30 min at r.t. in the presence or absence of probe **H5** (5 μM). Cells were washed twice in PBS 2% fetal calf serum (FCS) prior to collection. For microscopy, formalin-fixed paraffin-embedded slides of non-small cell lung cancer and paired non-cancerous lung were deparaffinized, rehydrated, stained with hematoxylin and eosin, followed by dehydration and imaging on a Zeiss AxioScan microscope.

**Reporting summary**. Further information on research design is available in the Nature Research Reporting Summary linked to this article.

## Data availability
Models were built with Maestro and AMBER16 using PDB id. 1IAU (https://www.ebi.ac.uk/pdbe/entry/pdb/1iau). The data from Figs. 1–4 and Supporting Info Fig. 1–19 of this study are available from the corresponding authors upon request. Source data are provided with this paper.

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

## Acknowledgements

J.I.S. and N.D.B. acknowledge funding from OPTIMA (EP/L016559/1). J.I.S. acknowledges funding from the Scottish Funding Council (H14052). L.M.T. acknowledges the support of Fundacion Antonio Martin Escudero (FAME) in the form of a post-doctoral fellowship. D.G. acknowledges funding from the Medical Research Council (MR/R01566X/1). T.K. acknowledges funding from an MRC Career Development Award (MR/S006982/1) and an MRC Centre Grant (MR/N022556/1). E.W.R. acknowledges funding from the Beatson Institute (A31287). J.J.J. and J.M. acknowledge funding from the Engineering and Physical Sciences Research Council (EP/P011330/1). N.O.C. acknowledges funding from Cancer Research UK (C42454/A28596) and the Brain Tumour Charity (GN-000676). A.R.A. acknowledges funding from a Cancer Research UK Clinician Scientist award (A24867). M.V. acknowledges funding from an ERC Consolidator Grant (DYNAFLUORS, 771443) and an ERC Proof of Concept Grant (957535). This project has received funding from the European Union's Horizon 2020 research and innovation program under the Marie Sklodowska-Curie grant agreement (859908). We thank the technical support from Tom G van Oorschot at the Radboud University Medical Centre, the QMRI Flow Cytometry and Confocal Advanced Light Microscopy facilities at the University of Edinburgh and Luxembourg Bio Technologies Ltd. (Rehovot) for the kind supply of reagents for peptide synthesis. We acknowledge biorender.com for assistance with figure creation.

## Author contributions

J.I.S. synthesized chemical compounds; J.I.S., L.M.T., D.G., E.J.T., and Z.C. performed in vitro experiments and analyzed data; N.D.B. performed cytospins and subsequent imaging; T.K. provided cell lines and assisted with in vitro experiments; A.B.-B. and E.W.R. performed time-lapse microscopy experiments; J.J.J. and J.M. performed computational studies; D.T. and A.S. designed and performed experiments in mouse preclinical models; L.M.T., J.D., N.O.C., and M.F. designed and performed phenotypic screens; B.P., I.J.V., M.V., R.A.C., and A.R.A. obtained samples, designed, and performed experiments with human biopsies; J.I.S. and M.V. conceived the project, analyzed and discussed the results, and wrote the paper with contributions from all authors; M.V. supervised the overall project.

## Competing interests

The University of Edinburgh has filed a patent covering some of the technology described in this manuscript. J.M. is a current member of the Scientific Advisory Board of Cresset. The remaining authors declare no competing interests.
