## [Peer Review File · Nature Communications]

REVIEWER COMMENTS

Reviewer #1 (Remarks to the Author):

The manuscript entitled A highly reactive fluorogenic probe for granzyme B enables in-biopsy evaluation and screening of anticancer immunotherapies has the moderate significance to the field of biological science. Within the manuscript, authors described the development of substrate-based probe for GrB detection and its utility for biological assay.

The manuscript is divided for two parts. A chemical and biological part. While the second one is in my opinion well designed, the first one lacks a lot of controls, and this may be influencing the proper analysis of biological results. Cancer cells are abundant with different proteases including MMPs, that can be responsible for majority of signal. Therefore, to be sure, that biological results are correct, extensive validation of the probe is necessary. For now, the work always do not support the conclusions, therefore the major revision is needed and the article may be considered for second revision if additional results will be provided.

Major points:

Recently, the paper concerning noninvasive detection of GrB by Janiszewski et al. was published in J. Biol. Chem. This group used not only IEPD as a scaffold, but also elongated the peptide chain and changed amino acids in the peptide-based probe for GrB to increase the selectivity and potency of this probe. As one of the advantages of Scott et al. probe, H5 authors pointed out the peptide chain elongation and its positive influence for GrB activity. This is nothing new, as it was done before. I do not see the clear explanation why did the Authors synthesize their probe and did not use the already published sequence? I found it crucial to compare results with this substrate.

Selectivity of H5 probe was tested only on GrB, inactive GrB, active GrA, active casp-3 and active HNE. Active GrA and HNE possess different substrate specificity while compare with GrB. Why did the authors use these enzymes? To test the selectivity other caspases should be applied, especially caspase-6 that shares similar specificity with GrB. Without this test it is hard to say if this probe is selective. Please provide the k_{cat}/K_m for H5 probe against other caspases (4, 5, 6, 7, 8, 9, 10) and MMPs. Because the probe is also used in mouse model, its specificity should be tested on mouse enzymes as well. I strongly encourage Authors to provide these results.

Page 6. Line 134. Why the inhibitor of GrB did not completely inhibit GrB? Why can we still observe the fluorescence response of H5 even while the inhibitor was applied?

Page 9, line 171-...The mouse model was used for real-time reinvigoration of T-cells in co-cultures with cancer cells, while the probe specificity was tested on human enzymes. That is weird, especially

when granzymes are analyzed. It is well-known that granzymes substrate specificity significantly differ between human and mouse enzymes.

It is not clear what is the mechanism of probe H5 delivery to the cells. That should be investigated. The cell permeability should be tested on all used cell lines.

The most important. Is the GrB inhibitor inhibiting the GrB activity within the cells? In all imaging and FL experiments, control with GrB inhibitor should be provided. Without this, it is hard to believe that the observed fluorescence is due to the cleavage only <https://woodforjoy.pl/sklep/skladany-kitchen-helper/by-GrB>, not by other enzymes like MMPs.

Minor points:

Page 5. Why was the peptide elongated only with P1' and P2'? Did the Authors consider peptide elongation with more amino acid residues? Please explain within the text.

Page 6. Line 119. Catalytic efficiency is indeed high. Do the authors think that donor-acceptor pair may influence the catalytic efficiency? Please explain.

Table 1 lack substrate probe from citation 27.

Figure S5 does not distinguish active and inactive enzyme as it was written at page 9 line 180.

Page 9, lines:181-182. Provided results do not indicate that expressed enzyme is inactive. At fig. 2b and S5 and S6 we can observe the increase of the GrB concentration, however, antibodies do not discriminate between active/inactive form.

Figure 2e – please provide the bright field of the cells, as these results are not convincing now.

Reviewer #2 (Remarks to the Author):

In the manuscript, the authors developed a highly reactive granzyme B-responsive probe that allowed more sensitive detection of granzyme B as an indication for cytotoxic T cell activity against tumor cells. The study focused on synthesizing and characterization of several different probes by altering some side amino acids attached to the IEPD substrate to find the optimal imaging probe. In vitro experiments were done as a proof-of-concept for the probe activity and validated in ex vivo assessments. The probe was used to screen multiple anticancer drugs that can promote T cell-

mediated cancer cell apoptosis. The FRET-based IEPD probe or variations of this IEPD probe have already been reported, which significantly diminishes the novelty (J. Biol. Chem. (2020) 295(28) 9567–9582; Sci. Adv. 2020; 6: eabc2777). Also, these probes have already been used to monitor Granzyme B levels as an indicator of T cell response after immunotherapy. Following are my concerns:

Major points:

1. Page 4 – line 79: A clear rationale for conjugating fluorophore at the N-terminal and the quencher next to the cleavage site is not provided.
2. Page 5 – line 91: What is the rationale behind starting with AG and SG, but not other combinations of amino acids, after aspartic-acid cleaving point on H1 and H2 hexapeptides? The choice of AG and SG should be justified.
3. Figure 2b,c, and lines 114 to 117: There is a difference in conclusions for these two figures where in figure 2b, it was reported that H5 was able to reach over 150 fold in fluorescence intensity after incubating with hGzmB. However, in figure 1c, with the same amount of H5 probe and hGzmB, there was 16-fold higher. The authors should clarify this.
4. Page 6 – lines 134 and 135: Figure 1d does not support the statement that by adding Ac-IEPD-CHO, the fluorescence activation of H5 was reduced. This statement differs from the figure legend 1d, where it shows concentration-dependent cleavage of H5 by granzyme B without any mention of Ac-IEPD-CHO.
5. Page 7 – line 150 to 155: The assumptions used to explain how the H5 binding mode is more efficient than others could use some reference.
6. Page 9 – line 174: Are these CD8+ T cells primed to attack E0771 tumor cells specifically? While later in this section, OT-I T cells were used to recognize and kill OVA-pulsed tumor cells but here, only CD8+ were mentioned without clarifying how these T cells are specific for E0771 cells.
7. Page 10 – line 199: What is the reasoning behind changing co-culture with E0771 mammary tumor cells to OVA-pulsed EL4 tumor cells? It was claimed on lines 199 – 200 that H5 was tested for real-time reporting for OVA-EL4 and OT-I co-culture, then was H5 able to report E0771 cell death events in real-time as well? The authors can explain this better so that readers will not be confused.

8. Page 1 – line 210 to 213: Is the experimental design in Figure S8 similar to experiments with OT-I T cells in Figure 2? How were the 1 nM increments measured? It would be helpful if the authors clearly explain the “mimicking aspects” that allowed “multiple contacts from CD8+”.

9. Figure 2d: While the H5 probe showed activation with the granzyme B enzyme alone, as shown in Figures 1b and 1c, the signal in 2d seems unclear? The contrast in the images in Fig. 2b looks different.

10. Figure 2b: It is unclear if the probe only measures granzyme b activity inside cancer cells or the granzyme B released in the synapse.

11. Figure 2b and d: How is the peptide probe uptaken by cancer cells, and how effectively?

12. Please provide quantification of the H5 signal obtained from Figure 2d.

13. Figure 3: The mouse model used showed a very high percentage of tumor cells death (~60%) and higher CD8+ T cells (~45%) in the tumor at 14 days. But the probe showed a significantly higher granzyme B signal in cancer cells (~35-40%). It is not clear what percentage of these cells are already dead or dying? If the cells are viable, do the majority of these cells have granzyme B activity?

14. Figure 3, the since the BODIPY-FL is used as a fluorophore; it significantly diminishes the utility of the probe in imaging the granzyme B activity in complex tissue samples, such as tumors. The probe might also find limited use in monitoring immunotherapy efficacy since the response is slower and the granzyme B levels are lower. These limitations of the imaging probe should be clearly mentioned.

15. Figure 3e: It would have been helpful to quantify the granzyme B levels in the tumor tissue and correlate it to the activity of the imaging probe.

16. Page 13 – line 246 to 248: Please explain how the probe was preferentially uptaken by tumor cells but not others. Previously, it was shown that probe signal was detected within tumor cells but not T cells when co-culture was performed. However, many other subsets of cells in the tumor microenvironment, especially macrophages, can also engulf substances around them. At-SNE plot with an “always-on” probe could benefit authors to see better where probes are distributed within the tumor, and this can further support that H5 only gets activated due to granzyme B.

17. Figure 4: The screen was performed using CD8+ T cells and E0771 cancer cells. It is unclear how specific the T cells are against the cancer cells (what antigen do the T cells recognize and bind to for inducing the cytotoxic activity).

18. Figure 4: PI3K/AKT and mTOR pathways have been shown to play an important role in T cell activation and functions. Specifically, PI3K activation in T cells promotes survival, cell cycle progression, modulates differentiation and controls acquisition of effector and memory phenotypes. So, the inhibitors of these pathways can interfere with T cell activation and functions. However, it is surprising that the AKT inhibitor was shown to enhance T cell-mediated granzyme B activity. Also, cytotoxic agents such as taxanes (Paclitaxel and Docetaxel with IC50 in subnanomolar but used at micromolar concentrations) were used in the screen and induced some granzyme B activity. Please justify.

19. Figure 4c: There was no explanation about the H&E staining result and how this is related to the outcome of H5 activity in later assays with flow cytometry. The authors can include some clear explanations for figure 4c.

Minor points:

20. Page 5 – line 102: A brief clarification on how arginine in P1' or P2' was used to disfavor the binding affinity can be helpful. Even though the authors later-on explained closer to the end of page 7, but readers may start wondering from line 102.

21. Page 10 – line 202: Please clarify what cells were counterstained with cell tracker orange.

Reviewer #3 (Remarks to the Author):

This is a nice piece of chemical biology paper by Professor Marc Vendrell et al., in that novel, highly sensitive and selective FRET-based fluorescence probes for granzyme B were designed, developed, and applied to the live imaging of T cell-mediated cancer cell death in co-cultured systems. Furthermore, the authors succeeded to show the utility of their probe for the screening of drugs capable of invigorating the killing capacity of T cells, and for the clinical characterization of human tumor biopsies. So, this reviewer fully supports its publication in Nature Communications, one of the most prestigious journals in an interdisciplinary research field.

However, for a better understanding of the behavior and capability of their probe, this reviewer would like to request some revisions, as follows.

First, as for the design of hexapeptide probes, the authors discussed P1' and P2' only from the viewpoint of binding to GzmB, i.e., K_m , but judging from the results in table 1, P1' might have a large effect on k_{cat} . Results of the consumption rate of peptides in table S2 and S/N ratios in fig. 1b could be found, but they are not enough for understanding the molecular design in detail. So, please provide k_{cat} and K_m values for other probes than H5, and discuss more for example why Ala@P1' could accelerate k_{cat} .

Second, as for the molecular mechanism and the mode of action of their probe to detect T cell-mediated cancer cell killing, it should be quite important to provide concrete results whether the probe H5 is membrane-permeable or not. H-probes are hexapeptide probes bearing two carboxylic groups but have two hydrophobic cores of BODIPY and Dabcyl, So, readers cannot see easily whether these probes are membrane-permeable, and if so, where to be accumulated in living cells. This reviewer would like to know also more details about the mode of action, i.e, where the probe reacts with GzmB to become fluorescent. GzmB is known to exocytosed from T cells, therefore there must be a chance to react in extracellular media to stain all cells in the dish which might cause a high background signal, but the authors claimed that cancer cells were selectively stained with apparent puncta in fig 2d or rather diffusively in fig 2e. Please discuss the details of molecular mechanisms to detect dead cancer cells by T cell-related GzmB, more extensively. Providing a cartoon of the proposed mechanism would be helpful.

Third, as for the selectivity, the authors provided the comparison with caspase-3 and GzmA in fig S3, but this reviewer would like to know whether H5 reacts with other caspases 7 and 9, because these enzymes are well known to be released in cytosol upon apoptosis. If the probe is membrane-permeable, results with staurosporine in fig 2c might be the answer for this question.

Fourth, the authors showed beautiful results of tumor regression in fig 3, but let me ask a simple question. In figure 3g, there found a big variety of RFU in the cancer region, left down cells were highly fluorescent, and right upper ones not fluorescent. What was the cause of this variety?

Lastly, in this reviewer's opinion, as for the application to human lung tumor tissues, these kinds of data are quite important as a high-impact interdisciplinary research paper. Let me ask another simple question; is there any possibility to obtain direct fluorescent images of GzmB-mediated killing of tumor cells? In this manuscript, the authors conducted rather complicated approaches with FACS, which cannot be easily done in the clinical situation, and more importantly, destructive analysis. Please discuss the feasibility of their approaches as a real clinical diagnostic means for personalized anticancer immunotherapies.

Minor point:

Figure S8; the authors stated that this is an in vitro experiment mimicking cancer cells contacted multiply by CD8+ T cell, but this reviewer could not understand what the RFU means. Is this the fluorescence intensity in cells? Please provide more details of experimental conditions.

Response to reviewers

Reviewer 1

Recently, the paper concerning noninvasive detection of GrB by Janiszewski et al. was published in J. Biol. Chem. This group used not only IEPD as a scaffold, but also elongated the peptide chain and changed amino acids in the peptide-based probe for GrB to increase the selectivity and potency of this probe. As one of the advantages of Scott et al. probe, H5 authors pointed out the peptide chain elongation and its positive influence for GrB activity. This is nothing new, as it was done before. I do not see the clear explanation why did the Authors synthesize their probe and did not use the already published sequence? I found it crucial to compare results with this substrate.

Answer: We thank the reviewer for the comment. We cited the work by Janiszewski et al (ref 27 in the manuscript) and we have now included the qTJ71 probe in the Table 1 of the revised manuscript. We did not use this sequence because we started the chemical development of H5 several years ago, when longer FRET-based GzmB peptide sequences had not been reported yet. On the other hand, the reactivity of probe H5 and its low limit of detection for GzmB are significantly better than those of any previously reported FRET probes (as summarised in Table 1), so this work is novel in terms of chemical probe optimisation and biological applications. Our peptide design was inspired by the discoveries of Rut et al (Biol. Chem., 2016; reference 40 in the revised manuscript) where hexapeptides showed significantly better reactivity than shorter sequences for chymotrypsin-like and caspase-like proteases.

Selectivity of H5 probe was tested only on GrB, inactive GrB, active GrA, active casp-3 and active HNE. Active GrA and HNE possess different substrate specificity while compare with GrB. Why did the authors use these enzymes? To test the selectivity other caspases should be applied, especially caspase-6 that shares similar specificity with GrB. Without this test it is hard to say if this probe is selective. Please provide the k_{cat}/K_m for H5 probe against other caspases (4, 5, 6, 7, 8, 9, 10) and MMPs. Because the probe is also used in mouse model, its specificity should be tested on mouse enzymes as well. I strongly encourage Authors to provide these results.

Answer: We thank the reviewer for this important point. We have remarkably extended our selectivity panel and now included results of the reactivity of probe H5 against GzmB, GzmA, neutrophil elastase, caspases (4, 5, 6, 7, 8, 9 and 10) and matrix metalloproteinases (MMP-2 and 9). The results confirmed the excellent selectivity of probe H5 for GzmB. Because the reactivity of probe H5 with all other enzymes was very low, we could not reliably determine the k_{cat} and K_M values of probe H5 for these enzymes. These results have now been included in the new Figure 1e.

Page 6. Line 134. Why the inhibitor of GrB did not completely inhibit GrB? Why can we still observe the fluorescence response of H5 even while the inhibitor was applied?

Answer: This is an interesting comment. In the experiments shown in Figure 1e, we used Ac-IEPD-CHO, a reversible commercial inhibitor (Abcam) that contains an aldehyde group as the reactive warhead. Because the reaction between the aldehyde group and serine residues at the active enzyme site is reversible, a small fraction of inhibited enzyme can get 'uncaged' over time and subsequently react with probe H5. We have clarified that Ac-IEPD-CHO is an aldehyde-based reversible inhibitor of GzmB in the revised manuscript.

Page 9, line 171-...The mouse model was used for real-time reinvigoration of T-cells in co-cultures with cancer cells, while the probe specificity was tested on human enzymes. That is

weird, especially when granzymes are analyzed. It is well-known that granzymes substrate specificity significantly differ between human and mouse enzymes.

Answer: We thank the reviewer for highlighting this point. We tested the reactivity of probe H5 against both mouse and human GzmB and we observed that probe H5 reacts with both enzymes (Figure 1 and Figure S4). As the reviewer points out, this result was unexpected because we had synthesized probe H5m (replacing Pro by Phe) as a potentially more reactive probe for mouse GzmB. When we compared the probes H5 and H5m against active recombinant mouse GzmB, we observed a higher fluorescence response for H5 (results shown Figure S4), therefore we decided to use probe H5 for all our biological studies either in mouse cells or in human cells.

It is not clear what is the mechanism of probe H5 delivery to the cells. That should be investigated. The cell permeability should be tested on all used cell lines.

Answer: In order to study the cell permeability of our probes, we have synthesized a new compound (H5-unquenched), which is the peptide fragment of probe H5 that is released upon reaction with GzmB. Compound H5-unquenched contains one molecule of BODIPY-FL and is always fluorescent, therefore it allowed us to monitor permeability in the different cancer cells that we used in this work (E0771, OVA-EL4 and SCC cells). Our flow cytometry results demonstrate that compound H5-unquenched can readily enter all the cancer cell lines tested. Because both compounds H5 and H5-unquenched are based on IEPD peptide sequences and they mainly differ in the presence or absence of quencher, these experiments suggest that probe H5 can enter target cells. We have included these results in the Supporting Information (new Figure S9).

The most important. Is the GrB inhibitor inhibiting the GrB activity within the cells? In all imaging and FL experiments, control with GrB inhibitor should be provided. Without this, it is hard to believe that the observed fluorescence is due to the cleavage only by GrB, not by other enzymes like MMPs.

Answer: Thank you to the reviewer for highlighting this issue. In our first version of the manuscript, we had already provided images of co-cultures of E0771 cells and murine CD8+ T cells without and with the Ac-IEPD-CHO GzmB inhibitor, which shows a markedly reduced signal under confocal microscopy (Figure 2d) which we quantified by image analysis (new Figure S7). The expanded selectivity data shown in the new Figure 1e also indicates that probe H5 cannot be cleaved by some MMPs (e.g., MMP-2 and 9, which play important roles in cancer) and therefore it is unlikely that the fluorescence signals detected in cells are due to MMP cleavage. In addition, we have validated our observations with new flow cytometry experiments where we observed a significant reduction of the fluorescence signals of probe H5 in co-cultures that had been incubated with the GzmB inhibitor Ac-IEPD-CHO (revised Figure S6).

Page 5. Why was the peptide elongated only with P1' and P2'? Did the Authors consider peptide elongation with more amino acid residues? Please explain within the text.

Answer: During our chemical design, we considered to extend the tetrapeptide IEPD with two additional amino acids because of previous examples of hexapeptide sequences being favoured as substrates for chymotrypsin-like and caspase-like proteases (ref 40 in the revised manuscript). We have clarified this point in the revised manuscript.

Page 6. Line 119. Catalytic efficiency is indeed high. Do the authors think that donor-acceptor pair may influence the catalytic efficiency? Please explain.

Answer: We agree with the reviewer that this is an important point. Our results with different IEPD-based FRET sequences suggest that the donor-acceptor pair can have a strong influence in the reactivity with GzmB. As presented in the Table S1, we chose BODIPY FL as the fluorophore and Dabcyl as the quencher because that donor-acceptor pair gave the highest reactivity among the different fluorophore-quencher pairs tested in the IEPD sequence (around 14% conversion compared to other FRET pairs, which all showed less than 5% conversion). Furthermore, our computational studies shown in Figures 1f and 1g indicate that Dabcyl can occupy an optimal binding site of the enzyme to favour reactivity.

Table 1 lack substrate probe from citation 27.

Answer: We have included the qTJ71 probe in the revised Table 1.

Figure S5 does not distinguish active and inactive enzyme as it was written at page 9 line 180.

Answer: Thank you to the reviewer for spotting this typo. The results of these antibody-based detection assays cannot distinguish between active and inactive GzmB so we have removed the word 'inactive' from the figure legend.

Page 9, lines:181-182. Provided results do not indicate that expressed enzyme is inactive. At fig. 2b and S5 and S6 we can observe the increase of the GrB concentration, however, antibodies do not discriminate between active/inactive form.

Answer: We thank the reviewer for the comment. We agree that the results presented in Figure 2b and Figure S5 do not discriminate between active and inactive GzmB and we have amended the legend of Figure S5. However, the purpose of these experiments was to identify the conditions that would induce T cells to express higher levels of GzmB as a hallmark of T cell activation. With regards to the results presented in Figure S6, we performed these experiments using probe H5 and therefore the fluorescence signals observed are due to the presence of active GzmB.

Figure 2e – please provide the bright field of the cells, as these results are not convincing now.

Answer: The acquisition of neat brightfield images of cell co-cultures under time-lapse microscopy is challenging due to the need for constant refocusing of the field of view. For that reason, the images in Figure 2e included two additional fluorescence markers (CellTracker Orange for T cells, Sytox Blue for dead cells) that would provide additional information on the location of T cells and the viability of cancer cells, respectively. We have attempted to acquire additional movies where we included the brightfield channel on top of two fluorescence channels, and we have uploaded one representative movie as the new Movie S2. We hope that the reviewer will appreciate the technical challenges associated with these experiments.

Reviewer 2

Page 4 – line 79: A clear rationale for conjugating fluorophore at the N-terminal and the quencher next to the cleavage site is not provided.

Answer: We thank the reviewer for the comment. We hypothesized that, if we included the fluorophore at the N-terminal, the fluorescent fragments resulting from the reaction with GzmB may get trapped inside cells as they contain negative charges from Asp and Glu residues. This feature would favour accumulation and retention of the fluorescence signals. During the revision, we have synthesized a new compound (H5-unquenched) that allowed us to validate this hypothesis. We observed bright fluorescence of cancer cells upon incubation with compound H5-unquenched and after washing, as shown in the new Figure S9. Furthermore, our molecular dynamic studies (Figure 1f) indicate that the Dabcyl group at the C-terminal end can occupy a binding site in the enzyme to favour reactivity. Altogether, our results suggest that the positioning of BODIPY FL at the N-terminal end and the Dabcyl group at the C-terminal are optimal for the performance of probe H5 as a GzmB substrate.

Page 5 – line 91: What is the rationale behind starting with AG and SG, but not other combinations of amino acids, after aspartic-acid cleaving point on H1 and H2 hexapeptides? The choice of AG and SG should be justified.

Answer: We started the construction of our library with AG and SG based on previous reports covering the substrate specificity of GzmB. Specifically, the proteomic study by Van Damme et al (Mol. Cell. Proteomics, 2009; reference 41 in the revised manuscript) highlighted the preference of A and S in the P1' position when analysing a collection of 300+ native GzmB substrates. In the study by Harris et al (J. Biol. Chem., 1998: reference 39 in the revised manuscript), a phage display library was used to study the positions P3, P1' and P2' of synthetic granzyme B substrates, where the authors found the P2' position was heavily favoured by glycine.

Figure 2b,c, and lines 114 to 117: There is a difference in conclusions for these two figures where in figure 2b, it was reported that H5 was able to reach over 150 fold in fluorescence intensity after incubating with hGzmB. However, in figure 1c, with the same amount of H5 probe and hGzmB, there was 16-fold higher. The authors should clarify this.

Answer: The different scales used on the y-axis of Figures 1c and 1e may be confusing. In Figure 1c, the y-axis are relative fluorescence units (RFU x 1000). We observed around 16,000 RFU for probe H5 after reacting with GzmB whereas probe H5 alone displayed RFU values around 100 (exact values provided in the Source Data file). Therefore, the fold fluorescence increases in both Figure 1c and 1e are similar.

Page 6 – lines 134 and 135: Figure 1d does not support the statement that by adding Ac-IEPD-CHO, the fluorescence activation of H5 was reduced. This statement differs from the figure legend 1d, where it shows concentration-dependent cleavage of H5 by granzyme B without any mention of Ac-IEPD-CHO.

Answer: We thank the reviewer for spotting this typo. We have amended the text in the revised manuscript and highlight Figure 1e instead of Figure 1d.

Page 7 – line 150 to 155: The assumptions used to explain how the H5 binding mode is more efficient than others could use some reference

Answer: We thank the reviewer for the comment. We have amended the text to clarify this point in the revised manuscript and added two more references (i.e., references 44 and 45). Briefly, we clarified that this binding mode fulfils the complementarity criteria that drives the

formation of densely packed hydrophobic structures found in prion proteins, and that mutations in residues that disrupt knobs-into-holes packing reduce the binding affinity of short peptides.

Page 9 – line 174: Are these CD8+ T cells primed to attack E0771 tumor cells specifically? While later in this section, OT-I T cells were used to recognize and kill OVA-pulsed tumor cells but here, only CD8+ were mentioned without clarifying how these T cells are specific for E0771 cells.

Answer: We thank the reviewer for these comments. The E0771/CD8+ T cell co-culture model utilises anti-CD3, anti-CD28 and IL-2 mediated activation to achieve T cell reinvigoration without the need for cell-specific antigen recognition. We adapted this assay from the work reported by Kitamura et al (Front. Immunol., 2018; reference 47 in the revised manuscript). With these experiments (results presented in Figure 2 and Figure S6), we demonstrated that probe H5 could detect GzmB activity in co-cultures where T cells had been reinvigorated (i.e., after IL-2 incubation for 24 h) but not in co-cultures where T cells remained quiescent and unable to kill cancer cells.

Page 10 – line 199: What is the reasoning behind changing co-culture with E0771 mammary tumor cells to OVA-pulsed EL4 tumor cells? It was claimed on lines 199 – 200 that H5 was tested for real-time reporting for OVA-EL4 and OT-I co-culture, then was H5 able to report E0771 cell death events in real-time as well? The authors can explain this better so that readers will not be confused.

Answer: We thank the reviewer for highlighting this point. After demonstrating that probe H5 can detect GzmB activity in E0771/CD8+ T cell co-cultures where T cells had been reinvigorated by IL-2 (Figure 2), we decided to examine whether probe H5 could also detect GzmB activity that was resulting from T cell mediated killing upon antigen-specific recognition. The OVA-EL4 and OT-I co-culture is an ideal system because it closely represents the killing response of CD8+ T cells in vivo. With regards to the real-time imaging, the kinetics of antigen-mediated cancer cell death are faster than non-specific killing owing to the extremely agonistic effect of the SIINFEKL OVA peptide on CD8+ T cell activation (as reported by Hogquist et al in Cell, 1994, 76, 17-27) and therefore more suitable for time-lapse microscopy experiments as we showed in Figure 2e and Movies S1 and S2. We have clarified these points in the revised manuscript.

Page 1 – line 210 to 213: Is the experimental design in Figure S8 similar to experiments with OT-I T cells in Figure 2? How were the 1 nM increments measured? It would be helpful if the authors clearly explain the “mimicking aspects” that allowed “multiple contacts from CD8+”.

Answer: We thank the reviewer for highlighting this issue and we have edited Figure S8 (now Figure S11) for clarity. The plot displays the fluorescence response of probe H5 upon addition of 1 nM increments of recombinant human GzmB over the course of 8 hours. The experiment was designed to show that probe H5 increases its fluorescence emission upon addition of small increments of GzmB since T cell-mediated killing requires multiple contacts between T cells and cancer cells (we added the new reference Weigelin et al, Nat. Commun., 2021; reference 48 in the revised manuscript). Because this experiment does not use any cells, we have removed the previous panel A from the figure.

Figure 2d: While the H5 probe showed activation with the granzyme B enzyme alone, as shown in Figures 1b and 1c, the signal in 2d seems unclear? The contrast in the images in Fig. 2b looks different.

Answer: The images shown in Figure 2b and Figure 2d are not comparable because they rely on different cell systems and imaging probes. In Figure 2b we used an anti-GzmB antibody in

T cells alone to label the total levels of GzmB expressed in T cells before and after invigoration. In Figure 2d, we used the E0771/CD8+ T cell co-culture system to label the active GzmB released into E0771 cancer cells upon attack by CD8+ T cells.

Figure 2b: It is unclear if the probe only measures granzyme b activity inside cancer cells or the granzyme B released in the synapse.

Answer: In this experiment, we used an antibody-based reagents (primary: rat anti-GzmB, secondary: goat anti-rat AF647-labelled antibody) but not the probe to compare the expression of GzmB in T cells before and after they were incubated with anti-CD3, anti-CD28 and IL-2. The results clearly showed that reinvigoration of T cells led to increased expression of GzmB in T cells, and therefore we transferred these conditions to the co-culture experiments with CD8+ T cells and E0771 cancer cells.

Figure 2b and d: How is the peptide probe uptaken by cancer cells, and how effectively?

Answer: We appreciate the comment from the reviewer. To study the cell permeability of our probes, we have synthesized the new compound H5-unquenched, which is the peptide fragment of probe H5 that is released upon reaction with GzmB. Compound H5-unquenched contains BODIPY-FL and is always fluorescent, therefore it allowed us to monitor permeability. Our flow cytometry results demonstrate that compound H5-unquenched can readily enter all the cancer cell lines tested (E0771, OVA-EL4 and SCC cells). Because the peptide sequences of the compounds H5 and H5-unquenched are the same, these experiments suggest that probe H5 can enter target cancer cells. We have included these results in the Supporting Information (new Figure S9).

Please provide quantification of the H5 signal obtained from Figure 2d.

Answer: We have analysed images and quantified the fluorescence intensity of probe H5 in this experiment. The results are now included in the new Figure S7.

Figure 3: The mouse model used showed a very high percentage of tumor cells death (~60%) and higher CD8+ T cells (~45%) in the tumor at 14 days. But the probe showed a significantly higher granzyme B signal in cancer cells (~35-40%). It is not clear what percentage of these cells are already dead or dying? If the cells are viable, do the majority of these cells have granzyme B activity?

Answer: We thank the reviewer for this insight. The results presented in Figure 3 where we analysed the fluorescence of probe H5 in SCC WT and SCC FAK (-/-) mouse tumours were obtained from Zombie NIR-negative live cells, as shown in the gating strategy presented in Figure S13. We have edited the titles of the y-axis in the plots in Figure 3c-f to clarify this point. Furthermore, the results in Figure 3e indicate that around 40% cancer cells of the SCC FAK (-/-) tumours display GzmB activity whereas only 5% show GzmB activity in SCC wild-type tumours.

Figure 3, the since the BODIPY-FL is used as a fluorophore; it significantly diminishes the utility of the probe in imaging the granzyme B activity in complex tissue samples, such as tumors. The probe might also find limited use in monitoring immunotherapy efficacy since the response is slower and the granzyme B levels are lower. These limitations of the imaging probe should be clearly mentioned.

Answer: We thank the reviewer for this comment. We agree that some of the optical properties of BODIPY FL (e.g., short excitation and emission wavelengths) may be limiting when it comes to in vivo imaging. Nevertheless, the data shown in Figure 4 confirms that probe H5 can be

used for ex vivo analysis of human tumour tissues. We have amended the text in the revised manuscript to clarify the limitations of BODIPY FL for in vivo imaging.

Figure 3e: It would have been helpful to quantify the granzyme B levels in the tumor tissue and correlate it to the activity of the imaging probe.

Answer: We thank the reviewer for the suggestion. We have performed new experiments where we have used ELISA assays to measure the levels of GzmB in both SCC wild-type and SCC FAK (-/-) tumours. The results confirmed that SCC FAK (-/-) tumours display significantly higher expression levels of GzmB when compared to SCC wild-type tumours, which correlates with the activity readouts obtained with probe H5. We have included this data in the new Figure S12.

Page 13 – line 246 to 248: Please explain how the probe was preferentially uptaken by tumor cells but not others. Previously, it was shown that probe signal was detected within tumor cells but not T cells when co-culture was performed. However, many other subsets of cells in the tumor microenvironment, especially macrophages, can also engulf substances around them. At-SNE plot with an “always-on” probe could benefit authors to see better where probes are distributed within the tumor, and this can further support that H5 only gets activated due to granzyme B.

Answer: We thank the reviewer for the comment. As mentioned above, our new experiments with the compound H5-unquenched suggest that these peptides can enter cells (new Figure S9) and we agree with the reviewer that different cells in the tumour microenvironment might take up the intact probe H5; however, fluorescence signals would be only detected when there is intracellular active GzmB to cleave the probe. Furthermore, in addition to the tSNE plots shown in Figures 3g and 3h, we have performed flow cytometry analysis of SCC FAK (-/-) mouse tumours for each of the isolated cell populations (e.g., cancer cells, CD45+ immune cells, monocytes/macrophages, CD4+ T cells and CD8+ T cells) and monitored their fluorescence signals following incubation with probe H5. These results are presented in the new Figure S13 and confirm that cancer cells displayed significantly higher fluorescence than any other cells, supporting the fact that higher levels of active GzmB are found in these cells.

Figure 4: The screen was performed using CD8+ T cells and E0771 cancer cells. It is unclear how specific the T cells are against the cancer cells (what antigen do the T cells recognize and bind to for inducing the cytotoxic activity).

Answer: As mentioned in a previous answer, the E0771/CD8+ T cell co-culture model utilises anti-CD3, anti-CD28 and IL-2 mediated activation to achieve T cell reinvigoration without the need for cell-specific antigen recognition. We adapted this assay from the work reported by Kitamura et al (Front. Immunol., 2018; reference 47 in the revised manuscript) and optimised it for the use of probe H5 as shown in Figure 2 and Figure S6.

Figure 4: PI3K/AKT and mTOR pathways have been shown to play an important role in T cell activation and functions. Specifically, PI3K activation in T cells promotes survival, cell cycle progression, modulates differentiation and controls acquisition of effector and memory phenotypes. So, the inhibitors of these pathways can interfere with T cell activation and functions. However, it is surprising that the AKT inhibitor was shown to enhance T cell-mediated granzyme B activity. Also, cytotoxic agents such as taxanes (Paclitaxel and Docetaxel with IC50 in subnanomolar but used at micromolar concentrations) were used in the screen and induced some granzyme B activity. Please justify.

Answer: We thank the reviewer for highlighting these points. Whilst we agree that some inhibitors could decrease T cell activation and thus reduce the release of active granzyme B,

work by Reid et al (Oncoimmunology, 2015; reference 50 in the revised manuscript) showed that the inhibition of the AKT pathway enhanced the proliferation and survival of CD8+ T cells and prolonged their cytokine and GzmB production ability. These observations are in line with our results found from the drug screening.

With regards to the working concentrations, we used these dosages on the basis of our previous experience in testing these compounds across multiple cancer cell line phenotypic assays. Because the screening considered only a single concentration per compound, we decided to use concentrations that would likely induce phenotypic responses in cells (e.g., cell growth, viability) without completely killing all cells.

Figure 4c: There was no explanation about the H&E staining result and how this is related to the outcome of H5 activity in later assays with flow cytometry. The authors can include some clear explanations for figure 4c.

Answer: The H&E images of non-cancerous and cancerous tissues shown in Figure 4c are representative histological images of the samples that were used for the analysis in Figure 4d. We decided to include them to feature some of the structural differences between cancerous and non-cancerous tissues. We have clarified this point in the revised manuscript.

Page 5 – line 102: A brief clarification on how arginine in P1' or P2' was used to disfavor the binding affinity can be helpful. Even though the authors later-on explained closer to the end of page 7, but readers may start wondering from line 102.

Answer: We have included a clarification in the revised manuscript (lines 17-19, page 5) to clarify that arginine residues in the positions P1' or P2' can disfavor binding.

Page 10 – line 202: Please clarify what cells were counterstained with cell tracker orange.

Answer: We have clarified in the revised manuscript that the T cells in Figure 2e and in the Movies S1 and S2 had been stained with Cell Tracker Orange.

Reviewer 3

As for the design of hexapeptide probes, the authors discussed P1' and P2' only from the viewpoint of binding to GzmB, i.e., K_m , but judging from the results in table 1, P1' might have a large effect on k_{cat} . Results of the consumption rate of peptides in table S2 and S/N ratios in fig. 1b could be found, but they are not enough for understanding the molecular design in detail. So, please provide k_{cat} and K_m values for other probes than H5, and discuss more for example why Ala@P1' could accelerate k_{cat} .

Answer: We thank the reviewer for the suggestion. We have further analysed the impact of the different amino acids in the P1' and P2' positions by determining the k_{cat} and k_{cat}/K_M values of the three most reactive compounds after probe H5 (i.e., compounds H2, H3 and H4). We have included these results in the new Table S4. In these experiments, we observed higher k_{cat} values when hydrophobic amino acids (Ala in H5 or Trp in H4) were found in the position P1' when compared to the more polar Ser (H2 or H3). Differences in the constants were also observed when variations happened at the P2' position (Gly for H2 vs Leu for H3). These observations align with the results of our molecular dynamic simulations (Figures 1f-g), where we could conclude that the alanine residue in P1' fills a small hydrophobic pocket in the catalytic cleft of GzmB and the tryptophan in P1' is too large to fit in the pocket. We have clarified these points in the revised version of the manuscript.

As for the molecular mechanism and the mode of action of their probe to detect T cell-mediated cancer cell killing, it should be quite important to provide concrete results whether the probe H5 is membrane-permeable or not. H-probes are hexapeptide probes bearing two carboxylic groups but have two hydrophobic cores of BODIPY and Dabcyl, So, readers cannot see easily whether these probes are membrane-permeable, and if so, where to be accumulated in living cells. This reviewer would like to know also more details about the mode of action, i.e, where the probe reacts with GzmB to become fluorescent. GzmB is known to exocytosed from T cells, therefore there must be a chance to react in extracellular media to stain all cells in the dish which might cause a high background signal, but the authors claimed that cancer cells were selectively stained with apparent puncta in fig 2d or rather diffusively in fig 2e. Please discuss the details of molecular mechanisms to detect dead cancer cells by T cell-related GzmB, more extensively. Providing a cartoon of the proposed mechanism would be helpful.

Answer: We thank the reviewer for the comments. To study the cell permeability of our probes, we have synthesized the compound H5-unquenched, which is the peptide fragment of probe H5 that is released upon reaction with GzmB. Compound H5-unquenched is always fluorescent and it was used to monitor cell permeability. Our flow cytometry results demonstrate that compound H5-unquenched is cell-permeable, which suggests that probe H5 could enter the target cancer cells. We agree with the reviewer than GzmB could be exocytosed to the extracellular medium, but we did not detect any extracellular signals in our fluorescence microscopy experiments (Figure 2e and Movies S1 and S2). Therefore, we believe that the most plausible mechanism to explain the mode of action of probe H5 is that inactive GzmB would be released from T cells and enter cancer cells (via perforin), where it would be activated and able to cleave probe H5. This mode of action would also be supported by our results in Figure 3g and the new Figure S13, where we observed much higher fluorescence signals in cancer cells than in any other cells found in the tumour microenvironment. In order to clarify this proposed mechanism, we have included a cartoon (new Figure S10), where we highlighted the molecular mechanisms at the immunological synapse between T cells and cancer cells.

As for the selectivity, the authors provided the comparison with caspase-3 and GzmA in fig S3, but this reviewer would like to know whether H5 reacts with other caspases 7 and 9,

because these enzymes are well known to be released in cytosol upon apoptosis. If the probe is membrane-permeable, results with staurosporine in fig 2c might be the answer for this question.

Answer: We thank the reviewer for this important point. We have extended our selectivity panel and now included results of the reactivity of probe H5 against GzmB, GzmA, neutrophil elastase, caspases (4, 5, 6, 7, 8, 9 and 10) and matrix metalloproteases (MMP-2 and 9). The results confirmed the excellent selectivity of probe H5 for GzmB and these results have now been included in the new Figure 1e.

The authors showed beautiful results of tumor regression in fig 3, but let me ask a simple question. In figure 3g, there found a big variety of RFU in the cancer region, left down cells were highly fluorescent, and right upper ones not fluorescent. What was the cause of this variety?

Answer: We agree with the reviewer that the staining of cancer cells in SCC FAK (-/-) tumours is heterogeneous, which is due to the fact that we chose a single timepoint (i.e., 14 days) for a consistent ex vivo analysis across models. In the SCC FAK (-/-) model, tumour regression can take up to 21 days; however, we decided to pick an earlier time point where the size of SCC wild-type and SCC FAK (-/-) tumours would be manageable (e.g., shorter time points would have led to very small SCC FAK (-/-) tumours whereas the size of SCC wild-type tumours would be too large at 21 days). Given the variability in T cell mediated killing, it would not be expected that all cancer cells would be undergoing T cell mediated killing at the same time, and this is reflected in the heterogeneity of probe H5 labelling among the cancer cells in Figure 3g.

In this reviewer's opinion, as for the application to human lung tumor tissues, these kinds of data are quite important as a high-impact interdisciplinary research paper. Let me ask another simple question; is there any possibility to obtain direct fluorescent images of GzmB-mediated killing of tumor cells? In this manuscript, the authors conducted rather complicated approaches with FACS, which cannot be easily done in the clinical situation, and more importantly, destructive analysis. Please discuss the feasibility of their approaches as a real clinical diagnostic means for personalized anticancer immunotherapies.

Answer: We appreciate the comment for the reviewer. Although flow cytometry is increasingly used in clinical settings to profile multiple cancer and immune cell populations in clinical specimens (e.g., blood, tissue biopsies), we agree with the reviewer that it has some shortcomings, such as limited spatial resolution. We have clarified this point in the revised manuscript and highlighted that further optimization will be required to optimise the application of probe H5 in fluorescence microscopy experiments using human biopsied tissues.

Figure S8; the authors stated that this is an in vitro experiment mimicking cancer cells contacted multiply by CD8+ T cell, but this reviewer could not understand what the RFU means. Is this the fluorescence intensity in cells? Please provide more details of experimental conditions.

Answer: We thank the reviewer for highlighting this issue and we have edited Figure S8 for clarity (now Figure S11). As we answered to Reviewer 2, the plot displays the fluorescence response of probe H5 upon addition of 1 nM increments of recombinant human GzmB over the course of 8 hours. The experiment was designed to show that probe H5 increases its fluorescence emission upon addition of small increments of GzmB since T cell-mediated killing requires multiple contacts between T cells and cancer cells. Because this experiment does not use any cells, we have removed the previous panel A from the figure.

REVIEWER COMMENTS

Reviewer #1 (Remarks to the Author):

Reviewer 1

Recently, the paper concerning noninvasive detection of GrB by Janiszewski et al. was published in J. Biol. Chem. This group used not only IEPD as a scaffold, but also elongated the peptide chain and changed amino acids in the peptide-based probe for GrB to increase the selectivity and potency of this probe. As one of the advantages of Scott et al. probe, H5 authors pointed out the peptide chain elongation and its positive influence for GrB activity. This is nothing new, as it was done before. I do not see the clear explanation why did the Authors synthesize their probe and did not use the already published sequence? I found it crucial to compare results with this substrate.

Answer: We thank the reviewer for the comment. We cited the work by Janiszewski et al (ref 27 in the manuscript) and we have now included the qTJ71 probe in the Table 1 of the revised manuscript. We did not use this sequence because we started the chemical development of H5 several years ago, when longer FRET-based GzmB peptide sequences had not been reported yet. On the other hand, the reactivity of probe H5 and its low limit of detection for GzmB are significantly better than those of any previously reported FRET probes (as summarised in Table 1), so this work is novel in terms of chemical probe optimisation and biological applications. Our peptide design was inspired by the discoveries of Rut et al (Biol. Chem., 2016; reference 40 in the revised manuscript) where hexapeptides showed significantly better reactivity than shorter sequences for chymotrypsin-like and caspase-like proteases.

R Answer: The 10-fold difference between H5 and the other probe for GrB activity detection is not significant considering the high kinetic parameters of both. In both cases the enzyme reaction with probe will be rapid and this difference is not significant for biological assays. Therefore I do not see the novelty in here.

Selectivity of H5 probe was tested only on GrB, inactive GrB, active GrA, active casp-3 and active HNE. Active GrA and HNE possess different substrate specificity while compare with GrB. Why did the authors use these enzymes? To test the selectivity other caspases should be applied, especially caspase-6 that shares similar specificity with GrB. Without this test it is hard to say if this probe is selective. Please provide the k_{cat}/K_m for H5 probe against other caspases (4, 5, 6, 7, 8, 9, 10) and MMPs. Because the probe is also used in mouse model, its specificity should be tested on mouse enzymes as well. I strongly encourage Authors to provide these results.

Answer: We thank the reviewer for this important point. We have remarkably extended our selectivity panel and now included results of the reactivity of probe H5 against GzmB, GzmA,

neutrophil elastase, caspases (4, 5, 6, 7, 8, 9 and 10) and matrix metalloproteinases (MMP-2 and 9). The results confirmed the excellent selectivity of probe H5 for GzmB. Because the reactivity of probe H5 with all other enzymes was very low, we could not reliably determine the k_{cat} and K_M values of probe H5 for these enzymes. These results have now been included in the new Figure 1e.

R Answer: OK

Page 6. Line 134. Why the inhibitor of GrB did not completely inhibit GrB? Why can we still observe the fluorescence response of H5 even while the inhibitor was applied?

Answer: This is an interesting comment. In the experiments shown in Figure 1e, we used AcIEPD-CHO, a reversible commercial inhibitor (Abcam) that contains an aldehyde group as the reactive warhead. Because the reaction between the aldehyde group and serine residues at the active enzyme site is reversible, a small fraction of inhibited enzyme can get 'uncaged' over time and subsequently react with probe H5. We have clarified that Ac-IEPD-CHO is an aldehyde-based reversible inhibitor of GzmB in the revised manuscript.

R Answer: There are other available irreversible inhibitors of GzmB. GrB is a serine protease so also the generic serine proteases irreversible inhibitors could be applied. In a case of this analysis, a few inhibitors should be tested as these data are not convincing. I am sorry. There are different % inhibition observed between experiments. For example, at Fig. 1e and Fig. S7 the inhibition of enzyme activity is around 90%, while results at Fig. S6 d indicates only 75% inhibition. Considering very low signal at figure 1e, I do not see it as a good control. This figure (1e) results should be presented/analyzed at different z-stacks, not only 1, and the quality of signal should be improved. Especially having such a bright tag in the probe, it should not be a problem. This experiment is crucial for the rest of the manuscript, as it this substrate is a basis for other experiments.

Page 9, line 171-...The mouse model was used for real-time reinvigoration of T-cells in cocultures with cancer cells, while the probe specificity was tested on human enzymes. That is 2 weird, especially when granzymes are analyzed. It is well-known that granzymes substrate specificity significantly differ between human and mouse enzymes.

Answer: We thank the reviewer for highlighting this point. We tested the reactivity of probe H5 against both mouse and human GzmB and we observed that probe H5 reacts with both enzymes (Figure 1 and Figure S4). As the reviewer points out, this result was unexpected because we had synthesized probe H5m (replacing Pro by Phe) as a potentially more reactive probe for mouse GzmB. When we compared the probes H5 and H5m against active recombinant mouse GzmB, we observed a higher fluorescence response for H5 (results shown Figure S4), therefore we decided to use probe H5 for all our biological studies either in mouse cells or in human cells.

R Answer: OK

It is not clear what is the mechanism of probe H5 delivery to the cells. That should be investigated. The cell permeability should be tested on all used cell lines.

Answer: In order to study the cell permeability of our probes, we have synthesized a new compound (H5-unquenched), which is the peptide fragment of probe H5 that is released upon reaction with GzmB. Compound H5-unquenched contains one molecule of BODIPY-FL and is always fluorescent, therefore it allowed us to monitor permeability in the different cancer cells that we used in this work (E0771, OVA-EL4 and SCC cells). Our flow cytometry results demonstrate that compound H5-unquenched can readily enter all the cancer cell lines tested. Because both compounds H5 and H5-unquenched are based on IEPD peptide sequences and they mainly differ in the presence or absence of quencher, these experiments suggest that probe H5 can enter target cells. We have included these results in the Supporting Information (new Figure S9).

R Answer: This experiment is wrongly designed. You cannot compare H5 uptake by the study of probe with different structure! H5-unquenched probe lacks, among others, quencher and therefore may differently interact with the cells. Other experiment should be provided.

The most important. Is the GrB inhibitor inhibiting the GrB activity within the cells? In all imaging and FL experiments, control with GrB inhibitor should be provided. Without this, it is hard to believe that the observed fluorescence is due to the cleavage only by GrB, not by other enzymes like MMPs.

Answer: Thank you to the reviewer for highlighting this issue. In our first version of the manuscript, we had already provided images of co-cultures of E0771 cells and murine CD8+ T cells without and with the Ac-IEPD-CHO GzmB inhibitor, which shows a markedly reduced signal under confocal microscopy (Figure 2d) which we quantified by image analysis (new Figure S7). The expanded selectivity data shown in the new Figure 1e also indicates that probe H5 cannot be cleaved by some MMPs (e.g., MMP-2 and 9, which play important roles in cancer) and therefore it is unlikely that the fluorescence signals detected in cells are due to MMP cleavage. In addition, we have validated our observations with new flow cytometry experiments where we observed a significant reduction of the fluorescence signals of probe H5 in co-cultures that had been incubated with the GzmB inhibitor Ac-IEPD-CHO (revised Figure S6).

R Answer: As mentioned before, the quality of figure 1e is too low and considering the cell nucleus shape, it seems like the cells were analyzed at different cross sections. I strongly recommend to provide results for different z-stacks. I also suggest to use irreversible inhibitor of GrB. Flow cytometry analyzes with and without inhibitor were taken only after 2h. Additional control after 24h should also been provided.

Page 5. Why was the peptide elongated only with P1' and P2'? Did the Authors consider peptide elongation with more amino acid residues? Please explain within the text.

Answer: During our chemical design, we considered to extend the tetrapeptide IEPD with two additional amino acids because of previous examples of hexapeptide sequences being favoured as substrates for chymotrypsin-like and caspase-like proteases (ref 40 in the revised manuscript). We have clarified this point in the revised manuscript.

R Answer: OK

Page 6. Line 119. Catalytic efficiency is indeed high. Do the authors think that donor-acceptor pair may influence the catalytic efficiency? Please explain. 3

Answer: We agree with the reviewer that this is an important point. Our results with different IEPD-based FRET sequences suggest that the donor-acceptor pair can have a strong influence in the reactivity with GzmB. As presented in the Table S1, we chose BODIPY FL as the fluorophore and Dabcyl as the quencher because that donor-acceptor pair gave the highest reactivity among the different fluorophore-quencher pairs tested in the IEPD sequence (around 14% conversion compared to other FRET pairs, which all showed less than 5% conversion). Furthermore, our computational studies shown in Figures 1f and 1g indicate that Dabcyl can occupy an optimal binding site of the enzyme to favour reactivity.

Table 1 lack substrate probe from citation 27. Answer: We have included the qTJ71 probe in the revised Table 1. Figure S5 does not distinguish active and inactive enzyme as it was written at page 9 line 180.

R Answer: OK

Answer: Thank you to the reviewer for spotting this typo. The results of these antibody-based detection assays cannot distinguish between active and inactive GzmB so we have removed the word 'inactive' from the figure legend.

Page 9, lines:181-182. Provided results do not indicate that expressed enzyme is inactive. At fig. 2b and S5 and S6 we can observe the increase of the GrB concentration, however, antibodies do not discriminate between active/inactive form.

Answer: We thank the reviewer for the comment. We agree that the results presented in Figure 2b and Figure S5 do not discriminate between active and inactive GzmB and we have amended the

legend of Figure S5. However, the purpose of these experiments was to identify the conditions that would induce T cells to express higher levels of GzmB as a hallmark of T cell activation. With regards to the results presented in Figure S6, we performed these experiments using probe H5 and therefore the fluorescence signals observed are due to the presence of active GzmB.

R Answer: OK

Figure 2e – please provide the bright field of the cells, as these results are not convincing now.

Answer: The acquisition of neat brightfield images of cell co-cultures under time-lapse microscopy is challenging due to the need for constant refocusing of the field of view. For that reason, the images in Figure 2e included two additional fluorescence markers (CellTracker Orange for T cells, Sytox Blue for dead cells) that would provide additional information on the location of T cells and the viability of cancer cells, respectively. We have attempted to acquire additional movies where we included the brightfield channel on top of two fluorescence channels, and we have uploaded one representative movie as the new Movie S2. We hope that the reviewer will appreciate the technical challenges associated with these experiments.

R Answer: I appreciate the effort. This would be easier to analyze different z-stacks from the same cell at the same time. That way the bright field and other analyzed factors can be easily seen.

Reviewer #2 (Remarks to the Author):

The authors have addressed my comments.

Response to reviewers (manuscript ID: NCOMMS-21-20944B)

Reviewer #1

1. Recently, the paper concerning noninvasive detection of GrB by Janiszewski et al. was published in J. Biol. Chem. This group used not only IEPD as a scaffold, but also elongated the peptide chain and changed amino acids in the peptide-based probe for GrB to increase the selectivity and potency of this probe. As one of the advantages of Scott et al. probe, H5 authors pointed out the peptide chain elongation and its positive influence for GrB activity. This is nothing new, as it was done before. I do not see the clear explanation why did the Authors synthesize their probe and did not use the already published sequence? I found it crucial to compare results with this substrate.

Answer: We thank the reviewer for the comment. We cited the work by Janiszewski et al (ref 27 in the manuscript) and we have now included the qTJ71 probe in the Table 1 of the revised manuscript. We did not use this sequence because we started the chemical development of H5 several years ago, when longer FRET-based GzmB peptide sequences had not been reported yet. On the other hand, the reactivity of probe H5 and its low limit of detection for GzmB are significantly better than those of any previously reported FRET probes (as summarised in Table 1), so this work is novel in terms of chemical probe optimisation and biological applications. Our peptide design was inspired by the discoveries of Rut et al (Biol. Chem., 2016; reference 40 in the revised manuscript) where hexapeptides showed significantly better reactivity than shorter sequences for chymotrypsin-like and caspase-like proteases.

R Answer: The 10-fold difference between H5 and the other probe for GrB activity detection is not significant considering the high kinetic parameters of both. In both cases the enzyme reaction with probe will be rapid and this difference is not significant for biological assays. Therefore I do not see the novelty in here.

Response: The work presented in this manuscript covers a breadth of experimental work with novelty in both chemistry and biology. With regards to chemistry, we have rationally designed a highly reactive FRET probe by stepwise optimisation of all the different components (i.e., fluorophore quencher pairs, spacers and peptide sequence) and also demonstrated the binding mode of probe H5 by molecular simulations, which not only explains its reactivity but also will facilitate the design and optimization of future probes and inhibitors. On the biological front, we have demonstrated that 1) GzmB activity is directly correlated to T-cell mediated regression in a FAK knock-out mouse model, which is relevant in cancer immunotherapy given the current evaluation of FAK inhibitors in combination with immune checkpoint inhibitors in clinical trials, 2) the suitability of probe H5 for high-throughput drug screens which led to the discovery of the AKT kinase inhibitor AZD5363 as a new small molecule with immune-mediated anticancer activity, and 3) the application of probe H5 for the direct evaluation of human tissue biopsies from lung cancer patients. To the best of our knowledge, none of these aspects have been reported before in the literature.

2. Page 6. Line 134. Why the inhibitor of GrB did not completely inhibit GrB? Why can we still observe the fluorescence response of H5 even while the inhibitor was applied?

Answer: This is an interesting comment. In the experiments shown in Figure 1e, we used AcIEPD-CHO, a reversible commercial inhibitor (Abcam) that contains an aldehyde group as the reactive warhead. Because the reaction between the aldehyde group and serine residues at the active enzyme site is reversible, a small fraction of inhibited enzyme can get ‘uncaged’ over time and subsequently

react with probe H5. We have clarified that Ac-IEPD-CHO is an aldehyde-based reversible inhibitor of GzmB in the revised manuscript.

R Answer: There are other available irreversible inhibitors of GzmB. GrB is a serine protease so also the generic serine proteases irreversible inhibitors could be applied. In a case of this analysis, a few inhibitors should be tested as these data are not convincing. I am sorry. There are different % inhibition observed between experiments. For example, at Fig. 1e and Fig. S7 the inhibition of enzyme activity is around 90%, while results at Fig. S6 d indicates only 75% inhibition. Considering very low signal at figure 1e, I do not see it as a good control. This figure (1e) results should be presented/analyzed at different z-stacks, not only 1, and the quality of signal should be improved. Especially having such a bright tag in the probe, it should not be a problem. This experiment is crucial for the rest of the manuscript, as it this substrate is a basis for other experiments.

Response: We thank the reviewer for the comment. We did most of our inhibition experiments with Ac-IEPD-CHO (a commercial reported GzmB inhibitor) because 1) it is more selective for GzmB than generic serine protease inhibitors and 2) the reviewer requested further proof of selectivity (which we did with an extended panel of recombinant enzymes in the revision) so using a non-selective protease inhibitor seems counterintuitive. Nevertheless, we did flow cytometry experiments and confirmed that the signal of probe H5 is also blocked with the generic serine protease inhibitor (AAD-CMK), which is presented below in the Figure 1 for Reviewers. We can include those results in the Supporting Information of the manuscript if the Editor considers it appropriate.

With regards to the quantification point, we would like to clarify that we have quantified signal inhibition by flow cytometry (Figure S6). Flow cytometry is the gold standard for quantitative fluorescence assays because it measures the signals from thousands of cells -in our case, we use 10,000 counting events for all our assays- instead of tens or hundreds of cells as measured by fluorescence microscopy. Fluorescence microscopy assays are mostly informative for intracellular distribution and spatial resolution but are not as reliable as flow cytometry when it comes to quantification. As the reviewer will understand from their experience, the differences between 75% and 90% inhibition are part of the experimental variation between assays that rely on different techniques and equipment (e.g., spectrophotometer vs flow cytometer or fluorescence microscope).

Figure 1 for Reviewers. Representative flow cytometry contour plots of co-cultures of E0771 cells (50,000 cells well⁻¹) and activated murine CD8⁺ T cells (200,000 cells well⁻¹) in the absence and presence of different protease inhibitors. Excitation/emission wavelengths: 488 nm/525 nm (for probe H5). Representative plots from independent experiments performed in triplicate.

3. It is not clear what is the mechanism of probe H5 delivery to the cells. That should be investigated. The cell permeability should be tested on all used cell lines.

Answer: In order to study the cell permeability of our probes, we have synthesized a new compound (H5-unquenched), which is the peptide fragment of probe H5 that is released upon reaction with GzmB. Compound H5-unquenched contains one molecule of BODIPY-FL and is always fluorescent, therefore it allowed us to monitor permeability in the different cancer cells that we used in this work (E0771, OVA-EL4 and SCC cells). Our flow cytometry results demonstrate that compound H5-unquenched can readily enter all the cancer cell lines tested. Because both compounds H5 and H5-unquenched are based on IEPD peptide sequences and they mainly differ in the presence or absence of quencher, these experiments suggest that probe H5 can enter target cells. We have included these results in the Supporting Information (new Figure S9).

R Answer: This experiment is wrongly designed. You cannot compare H5 uptake by the study of probe with different structure! H5-unquenched probe lacks, among others, quencher and therefore may differently interact with the cells. Other experiment should be provided.

Response: We respectfully disagree with the reviewer as we think that this is the closest/best experiment we could do in live intact cells to assess the permeability of our probes (no suggestion was provided either). Indirect assays (e.g., HPLC, mass spectrometry) to detect intact probe H5 in cells are technically extremely challenging and involve protocols that either lyse cells or affect their membrane integrity, therefore defeating the purpose of this experiment.

We agree with the reviewer that probe H5-unquenched and probe H5 differ in the presence of the quencher but the extremely low fluorescence quantum yield of the intact FRET probe H5 -as it happens with many other FRET probes- makes it impossible to detect it before reaction with GzmB. Moreover, the probe H5-unquenched contains the whole peptide sequence -including the most impermeable chemical components of probe H5 (i.e., the two negatively charged residues Asp and Glu)- and therefore it is representative of how the whole intact FRET probes can enter cells. The main chemical difference is the presence/absence of the Dabcyl quencher group, which is hydrophobic and uncharged, and therefore unlikely to have major influence on the cell permeability of the peptide probes. Furthermore, we have performed our permeability analysis in all the different cancer cells that were used in this work (i.e., E0771, OVA-EL4 and SCC cells) and observed similar trends in all of them.

4. The most important. Is the GrB inhibitor inhibiting the GrB activity within the cells? In all imaging and FL experiments, control with GrB inhibitor should be provided. Without this, it is hard to believe that the observed fluorescence is due to the cleavage only by GrB, not by other enzymes like MMPs.

Answer: Thank you to the reviewer for highlighting this issue. In our first version of the manuscript, we had already provided images of co-cultures of E0771 cells and murine CD8⁺ T cells without and with the Ac-IEPD-CHO GzmB inhibitor, which shows a markedly reduced signal under confocal microscopy (Figure 2d) which we quantified by image analysis (new Figure S7). The expanded selectivity data shown in the new Figure 1e also indicates that probe H5 cannot be cleaved by some MMPs (e.g., MMP-2 and 9, which play important roles in cancer) and therefore it is unlikely that the fluorescence signals detected in cells are due to MMP cleavage. In addition, we have validated our observations with new flow cytometry experiments where we observed a significant reduction of the fluorescence signals of probe H5 in co-cultures that had been incubated with the GzmB inhibitor Ac-IEPD-CHO (revised Figure S6).

R Answer: As mentioned before, the quality of figure 1e is too low and considering the cell nucleus shape, it seems like the cells were analyzed at different cross sections. I strongly recommend to provide results for different z-stacks. I also suggest to use irreversible inhibitor of GrB. Flow cytometry

analyzes with and without inhibitor were taken only after 2h. Additional control after 24h should also been provided.

Response: The images shown in Figure 2d are representative and the quantification of fluorescence microscopy images was performed covering more than 10 different regions of interest (as stated in the figure legend). However, as mentioned previously, the fluorescence emission signals of probe H5 were properly quantified in our flow cytometry experiments (Figure S6), where 10,000 events were acquired for every population of interest (stated in the Reporting summary). With regards to the time point, we observed a declined in viability when cells had been incubated with protease inhibitors for 24 h so we used the 2 h time point for these experiments.

5. Figure 2e – please provide the bright field of the cells, as these results are not convincing now.

Answer: The acquisition of neat brightfield images of cell co-cultures under time-lapse microscopy is challenging due to the need for constant refocusing of the field of view. For that reason, the images in Figure 2e included two additional fluorescence markers (CellTracker Orange for T cells, Sytox Blue for dead cells) that would provide additional information on the location of T cells and the viability of cancer cells, respectively. We have attempted to acquire additional movies where we included the brightfield channel on top of two fluorescence channels, and we have uploaded one representative movie as the new Movie S2. We hope that the reviewer will appreciate the technical challenges associated with these experiments.

R Answer: I appreciate the effort. This would be easier to analyze different z-stacks from the same cell at the same time. That way the bright field and other analyzed factors can be easily seen.

Response: We thank the reviewer for understanding the technical challenges associated with these experiments. The time-lapse microscopy experiments shown in Figure 2e and Movies S1 and S2 were performed in a fluorescence epifluorescence microscope -rather than a confocal microscope- to be able to capture the movement of T cells and cancer cells in multiple directions and at the same time retain high temporal resolution. It would not be possible to obtain Z-stacks under these conditions.

Reviewer #4

1. Include BODIPY spectral information.

Answer: We have included a new figure in the revised Supporting Information (new Figure S3) which shows the absorbance and emission spectra as well as the fluorescence quantum yields for probe H5 and probe H5-unquenched.

2. Clarify the focus on IL2.

Answer: We have edited the manuscript to include a clarification that IL2 is an essential cytokine for the survival, proliferation and activation of CD8⁺ T cells (page 9).

3. Add a discussion of the applications of your approach.

Answer: We have extended the Conclusions section to include a more detailed discussion -as well as 5 new references- on different applications that could be found for probe **H5** as a selective fluorescent probe for active GzmB (pages 17 and 18).

4. Pre-treatment that your lung cancer patients had received.

Answer: The manuscript already mentioned that the results presented in Figure 4 correspond to tissue resections from treatment-naïve lung cancer patients undergoing surgical resection (page 15).